# Variational consistent histories as a hybrid algorithm for quantum foundations

Andrew Arrasmith [1,2], Lukasz Cincio[1], Andrew T. Sornborger[3], Wojciech H. Zurek[1] & Patrick J. Coles[1]

Although quantum computers are predicted to have many commercial applications, less attention has been given to their potential for resolving foundational issues in quantum mechanics. Here we focus on quantum computers' utility for the Consistent Histories formalism, which has previously been employed to study quantum cosmology, quantum paradoxes, and the quantum-to-classical transition. We present a variational hybrid quantum-classical algorithm for finding consistent histories, which should revitalize interest in this formalism by allowing classically impossible calculations to be performed. In our algorithm, the quantum computer evaluates the decoherence functional (with exponential speedup in both the number of qubits and the number of times in the history) and a classical optimizer adjusts the history parameters to improve consistency. We implement our algorithm on a cloud quantum computer to find consistent histories for a spin in a magnetic field and on a simulator to observe the emergence of classicality for a chiral molecule.

---

[1] Theoretical Division, MS 213, Los Alamos National Laboratory, Los Alamos, NM 87545, USA. [2] Department of Physics, University of California Davis, Davis, CA 95616, USA. [3] Information Sciences, MS 256, Los Alamos National Laboratory, Los Alamos, NM 87545, USA. Correspondence and requests for materials should be addressed to P.J.C. (email: pcoles@lanl.gov)

The foundations of quantum mechanics (QM) have been debated for the past century[1,2], including topics such as the Einstein–Podolsky–Rosen (EPR) paradox, hidden-variable theories, Bell's Theorem, Born's rule, and the role of measurements in QM. This also includes the quantum-to-classical transition, i.e., the emergence of classical behavior (objectivity, irreversibility, lack of interference, etc.) from quantum laws[3–5].

The Consistent Histories (CH) formalism was introduced by Griffiths[6], Omnès[7], Gell-Mann, and Hartle to address some (though not all) of the aforementioned issues[8]. One inventor considered CH to be "the Copenhagen interpretation done right"[6], as it resolves some of the paradoxes of QM by enforcing strict rules for logical reasoning with quantum systems. In this formalism, the Copenhagen interpretation's focus on measurements as the origin of probabilities is replaced by probabilities for sequences of events (histories) to occur, and hence by avoiding measurements it avoids the measurement problem. The sets of histories whose probabilities are additive (as the histories do not interfere with each other) are considered to be consistent and are thus the only ones able to be reasoned about in terms of classical probability and logic[7].

Regardless of one's opinion of the philosophical interpretation (on which this paper is agnostic), this computational framework has proven useful in applications such as attempting to solve the cosmological measure problem[9,10], understanding quantum jumps[11], and evaluating the arrival time for particles at a detector[12–14]. One of the main reasons that this framework has not received more attention and use is that carrying out the calculations for non-trivial cases (e.g., discrete systems of appreciable size or continuous systems that do not admit approximate descriptions by exactly solvable path integrals) can be difficult[11,15]. Although numerical approaches have been attempted[16,17], they require exponentially scaling resources as either the number of times considered or the system size grows. This makes classical numerical approaches unusable for any but the simplest cases.

With the impending arrival of the first noisy intermediate-scale quantum computers[18], the field of variational hybrid quantum-classical algorithms (VHQCAs), which make the most of short quantum circuits combined with classical optimizers, has been taking off. VHQCAs have now been demonstrated for a myriad of tasks ranging from factoring to finding ground states, among others[19–26]. The VHQCA framework potentially brings the practical applications of quantum computers years closer to fruition.

Here we present a scalable VHQCA for the CH formalism. Our algorithm achieves an exponential speedup over classical methods both in terms of the system size and the number of times considered. It will allow exploration beyond toy models, such as the quantum-to-classical transition in mesoscopic quantum systems. We implement this algorithm on IBM's superconducting qubit quantum processor and obtain results in good agreement with theoretical expectations, suggesting that useful implementations of our algorithm may be feasible on near-term quantum devices.

## Results

**Consistent histories background.** In the CH framework[27–29], a history $\mathcal{Y}^{\boldsymbol{\alpha}}$ is a sequence of properties (i.e., projectors onto the appropriate subspaces) at a succession of times $t_1 < t_2 < \ldots < t_k$,

$$\mathcal{Y}^{\boldsymbol{\alpha}} = (P_1^{\alpha_1}, P_2^{\alpha_2}, \ldots, P_k^{\alpha_k}), \tag{1}$$

where $P_j^{\alpha_j}$ is chosen from a set $P_j$ of projectors that sum to the identity at time $t_j$. For example, for a photon passing through a sequence of diffraction gratings and then striking a screen, a history could be the photon passed through one slit in the first grating, another slit in the second, and so on. Clearly, we find interference between such histories unless there is some sense in which the photon's path has been recorded. As there is interference, we cannot add the probabilities of the different histories classically and expect to correctly predict where the photon strikes the screen.

The CH framework provides tools for determining when a family (i.e., a set that sums to the multi-time identity operator) of histories $\mathcal{F} = \{\mathcal{Y}^{\boldsymbol{\alpha}}\}$ exhibits interference, which is not always obvious. In this framework, one defines the so-called class operator

$$\mathcal{C}^{\boldsymbol{\alpha}} = P_k^{\alpha_k}(t_k) P_{k-1}^{\alpha_{k-1}}(t_{k-1}) \ldots P_1^{\alpha_1}(t_1), \tag{2}$$

which is the time-ordered product of the projection operators (now in the Heisenberg picture and hence explicitly time dependent) in history $\mathcal{Y}^{\boldsymbol{\alpha}}$. If the system is initially described by a density matrix $\rho$, the degree of interference or overlap between histories $\mathcal{Y}^{\boldsymbol{\alpha}}$ and $\mathcal{Y}^{\boldsymbol{\alpha}'}$ is

$$\mathcal{D}(\boldsymbol{\alpha}, \boldsymbol{\alpha}') = \mathrm{Tr}(\mathcal{C}^{\boldsymbol{\alpha}} \rho \, \mathcal{C}^{\boldsymbol{\alpha}'\dagger}). \tag{3}$$

This quantity is called the decoherence functional. The consistency condition for a family of histories $\mathcal{F}$ is then

$$\mathrm{Re}(\mathcal{D}(\boldsymbol{\alpha}, \boldsymbol{\alpha}')) = 0, \quad \forall \boldsymbol{\alpha} \neq \boldsymbol{\alpha}'. \tag{4}$$

If and only if this condition holds do we say that $\mathcal{D}(\boldsymbol{\alpha}, \boldsymbol{\alpha})$ is the probability for history $\mathcal{Y}^{\boldsymbol{\alpha}}$. For computational convenience, we will instead work with a stronger condition[28]:

$$\mathcal{D}(\boldsymbol{\alpha}, \boldsymbol{\alpha}') = 0, \quad \forall \boldsymbol{\alpha} \neq \boldsymbol{\alpha}', \tag{5}$$

As we are presenting a numerical algorithm, it will also be useful to consider approximate consistency, where we merely insist that the interference is small in the following sense:

$$|\mathcal{D}(\boldsymbol{\alpha}, \boldsymbol{\alpha}')|^2 \leq \varepsilon^2 \mathcal{D}(\boldsymbol{\alpha}, \boldsymbol{\alpha}) \mathcal{D}(\boldsymbol{\alpha}', \boldsymbol{\alpha}'), \quad \forall \boldsymbol{\alpha} \neq \boldsymbol{\alpha}', \tag{6}$$

which guarantees that probability sum rules for $\mathcal{F}$ are satisfied within an error of $\varepsilon$[30].

To study consistency arising purely from decoherence (i.e., records in the environment), researchers have proposed a functional that instead takes a partial trace over E, which is (a subsystem of) the environment[31,32]:

$$\mathcal{D}_{\mathrm{pt}}(\boldsymbol{\alpha}, \boldsymbol{\alpha}') = \mathrm{Tr}_{\mathrm{E}}(\mathcal{C}^{\boldsymbol{\alpha}} \rho \, \mathcal{C}^{\boldsymbol{\alpha}'\dagger}). \tag{7}$$

With this modification, the consistency condition is

$$\mathcal{D}_{\mathrm{pt}}(\boldsymbol{\alpha}, \boldsymbol{\alpha}') = \mathbf{0}, \quad \forall \boldsymbol{\alpha} \neq \boldsymbol{\alpha}', \tag{8}$$

where $\mathbf{0}$ is the zero matrix. Instead of only signifying the lack of interference, partial-trace consistency singles out whether or not the records of the histories in the environment interfere. It is noteworthy that the full-trace condition of Eq. (5) is satisfied when this partial-trace consistency is satisfied, but the converse does not hold[31].

With this formalism in hand, we can now see why classical numerical schemes for CH have faced difficulty. For example, consider histories of a collection of $n$ spin-1/2 particles for $k$ time steps, depicted in Fig. 1. The number of histories is $2^{nk}$ and hence there are $\sim 2^{2nk}$ decoherence functional elements. Furthermore, evaluating each decoherence functional element $\mathcal{D}(\boldsymbol{\alpha}, \boldsymbol{\alpha}')$ requires the equivalent of a Hamiltonian simulation of the system, i.e., the multiplication of $2^n \times 2^n$ matrices. This means modern clusters would take centuries to evaluate the consistency of a family of histories with $k = 2$ time steps and $n = 10$ spins. Given this limitation, we can see why, for the most part, only toy models have been analyzed in this framework thus far.

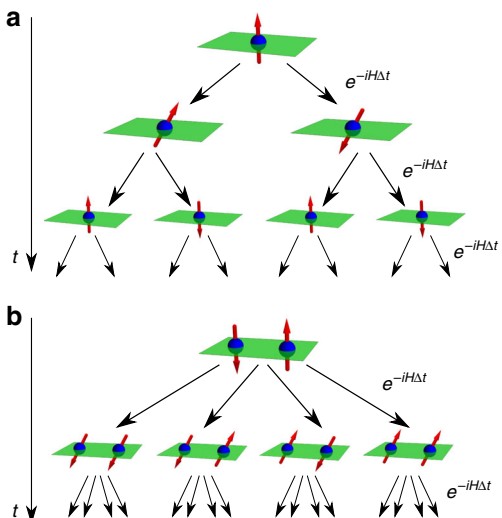

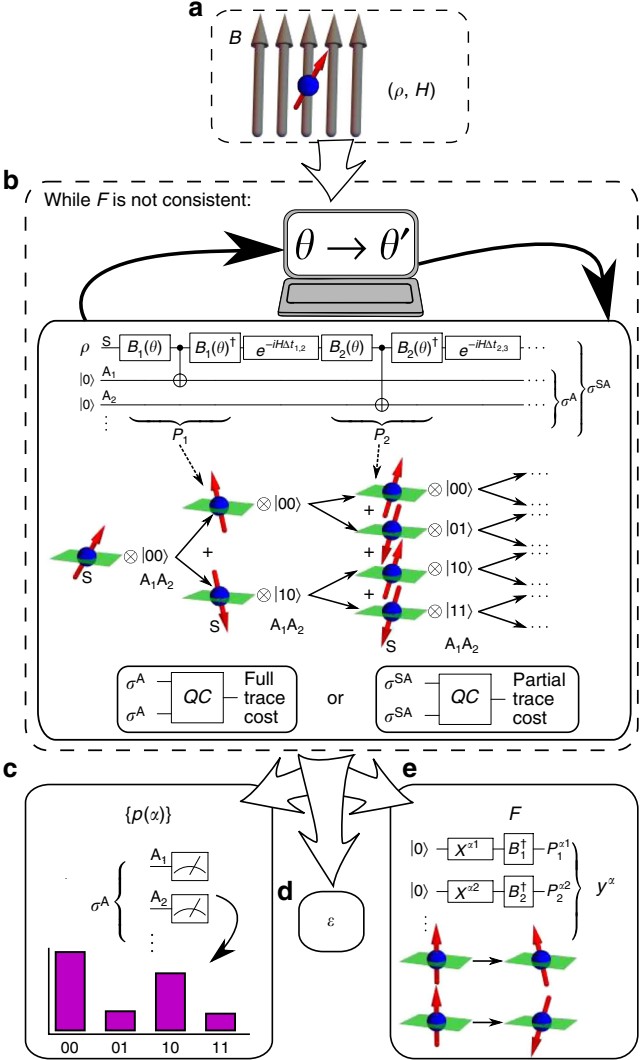

**Fig. 1** An illustration of the branching of histories for $k$ time steps. A one-spin ($n = 1$) and two-spin ($n = 2$) system, respectively, shown in **a**, **b**, have $2^k$ and $2^{2k}$ different histories. Here, $k = 3$ in **a** and $k = 2$ in **b**

**Hybrid algorithm for finding consistent histories**. We refer to our VHQCA as Variational Consistent Histories (VCH), see Fig. 2. VCH takes as its inputs a physical model (i.e., an initial state $\rho$ and a Hamiltonian $H$) and some ansatz for the types of projectors to consider. It outputs the following: (1) a family $\mathcal{F}$ of histories that is (approximately) full and/or partial-trace consistent in the form of projection operators prepared on a quantum computer, (2) the probabilities of the most likely histories $\mathcal{Y}^{\boldsymbol{\alpha}}$ in $\mathcal{F}$, and (3) a bound on the consistency parameter $\varepsilon$.

VCH involves a parameter optimization loop, where a quantum computer evaluates a cost function that quantifies the family's inconsistency, while a classical optimizer adjusts the family (i.e., varies the projector parameters) to reduce the cost. Classical optimizers for VHQCAs are actively being investigated[26,33] and one is free to choose the classical optimizer on an empirical basis.

To compute the cost, it is noteworthy that the elements of the decoherence functional form a positive semi-definite matrix with trace one. In VCH, we exploit this property to encode $\mathcal{D}$ in a quantum state $\sigma^{\mathrm{A}}$, whose matrix elements are $\langle \boldsymbol{\alpha} | \sigma^{\mathrm{A}} | \boldsymbol{\alpha}' \rangle = \mathcal{D}(\boldsymbol{\alpha}, \boldsymbol{\alpha}')$. Step **b** of Fig. 2 shows a quantum circuit that prepares $\sigma^{\mathrm{A}}$ (see Supplementary Note 2 for more details). This circuit transforms an initial state $\rho \otimes |\mathbf{0}\rangle\langle\mathbf{0}|$ on systems SA, where S simulates the physical system of interest and A is an ancilla system, into a state $\sigma^{\mathrm{SA}}$ whose marginal is $\sigma^{\mathrm{A}}$. For the full-trace consistency, we introduce a global measure of the (in) consistency that quantifies how far $\sigma^{\mathrm{A}}$ is from being diagonal, which serves as our cost function:

$$C := \sum_{\boldsymbol{\alpha} \neq \boldsymbol{\alpha}'} |\mathcal{D}(\boldsymbol{\alpha}, \boldsymbol{\alpha}')|^2 = D_{\mathrm{HS}}(\sigma^{\mathrm{A}}, \mathcal{Z}^{\mathrm{A}}(\sigma^{\mathrm{A}})), \quad (9)$$

where $D_{\mathrm{HS}}$ is the Hilbert–Schmidt distance and $\mathcal{Z}^{\mathrm{A}}(\sigma^{\mathrm{A}})$ is the dephased (all off-diagonal elements set to zero) version of $\sigma^{\mathrm{A}}$. This quantity goes to zero if and only if $\mathcal{F}$ is consistent. For the partial-trace case, we arrive at a similar cost function but with $\sigma^{\mathrm{A}}$ replaced by $\sigma^{\mathrm{SA}}$:

$$C_{\mathrm{pt}} := \sum_{\boldsymbol{\alpha} \neq \boldsymbol{\alpha}'} \left\| \mathcal{D}_{\mathrm{pt}}(\boldsymbol{\alpha}, \boldsymbol{\alpha}') \right\|_{\mathrm{HS}}^2 = D_{\mathrm{HS}}(\sigma^{\mathrm{SA}}, \mathcal{Z}^{\mathrm{A}}(\sigma^{\mathrm{SA}})). \quad (10)$$

Here, the notation $\mathcal{Z}^{\mathrm{A}}(\sigma^{\mathrm{SA}})$ indicates that the dephasing operation only acts on system A and the absolute squares of Eq. (9) have been generalized to Hilbert–Schmidt norms,

**Fig. 2** Flowchart for VCH. The goal of VCH is to take a physical model (**a**) and output an approximately consistent family $\mathcal{F}$ of histories (**e**), their associated probabilities $\{p(\alpha)\}$ (**c**), and a measure $\varepsilon$ of how consistent $\mathcal{F}$ is in (**d**). This is accomplished via a parameter optimization loop (**b**), which is a hybrid quantum-classical computation. Here the classical computer adjusts the projector parameters (contained in the gates $\{B_j(\theta)\}$, where $B_j(\theta)$ diagonalizes the $P_j$ projectors) and a quantum computer returns the cost. It is noteworthy that $P_j$ denotes the set of Schrodinger-picture projectors at the $j^{\mathrm{th}}$ time. The optimal parameters are then used to compute the probabilities of the most likely histories in $\mathcal{F}$ (**c**) and to prepare the projectors for any history in $\mathcal{F}$ (**e**, where $X$ is the Pauli-$X$ operator). Although the quantum circuits are depicted for a one-qubit system, the Supplementary Note 1 discusses the generalizations to multi-qubit systems, non-trivial environment E, coarse-grained histories, and branch-dependent histories

$\|M\|_{\mathrm{HS}}^2 := \mathrm{Tr}(M^{\dagger}M)$. In the Methods section, we present quantum circuits that compute these cost functions from two copies of $\sigma^{\mathrm{A}}$ or $\sigma^{\mathrm{SA}}$. Derivations of the second equalities in Eqs (9) and (10) can be found in Supplementary Note 3. We remark that alternative cost functions may be useful, e.g., to penalize families $\mathcal{F}$ with high entropy (see Methods) or to obtain a larger cost gradient by employing local instead of global observables (see ref. [26]).

The parameter optimization loop results in an approximately consistent family, $\mathcal{F}$, of histories, where the consistency parameter

$\varepsilon$ is upper bounded in terms of the final cost (see Methods). In Step **c** in Fig. 2, we then generate the probabilities for the most likely histories by repeatedly preparing $\sigma^A$ and measuring in the standard basis, where the measurement frequencies give the probabilities (an alternative circuit that reads out any one of the exponentially many elements $\mathcal{D}(\boldsymbol{\alpha}, \boldsymbol{\alpha}')$ is introduced in Supplementary Note 4). Step **e** shows how one prepares the set of projection operators for any given history in $\mathcal{F}$. These projectors can then be characterized with an efficient number of observables (i.e., avoiding full state tomography) to learn important information about the histories.

Let us discuss the scaling of VCH. With the potential exceptions of the Hamiltonian evolution and the projection operators, the complexity of our quantum circuits (i.e., the gate count, circuit depth, and total number of required qubits) scales linearly with both the system size $n$ and the number of times $k$ considered. The complexity of Hamiltonian evolution to some accuracy is problem dependent, but we typically expect polynomial scaling in $n$ for physical systems with properties like translational symmetry[34]. On the other hand, we consider the circuit depth for preparing the history projectors to be a refinement parameter. One can begin with a short-depth ansatz for the projectors and incrementally increase the depth to refine the ansatz, potentially improving the approximate consistency. We therefore expect the overall scaling of our quantum circuits to be polynomial in $n$ and $k$ for the anticipated use cases of VCH.

The complexity of minimizing our non-convex cost function is unknown, which is typical for VHQCAs. As classical methods for finding consistent families also involve optimizing over some parameterization for the projectors, classical methods also need to deal with this optimization complexity issue.

Although the number of required repetitions of the probability readout step can scale inefficiently in $n$ and $k$ for certain families of histories, we assume that minimizing the cost outputs a family $\mathcal{F}$ for which the probability readout step is efficient (see Methods for elaboration on this point).

This scaling behavior means that for systems that can be tractably simulated on a quantum computer and whose properties of interest are simple to implement, we achieve an exponential speedup and reduction in the needed resources as compared with classical approaches to this problem.

**Experimental implementations**. *Spin in a magnetic field*. We now present an experimental demonstration of VCH on a cloud quantum computer. See the Supplementary Note 5 for further details on this implementation. We examine the two time histories of a spin-1/2 particle in a magnetic field $B\hat{z}$, whose Hamiltonian is $H = -\gamma B\sigma_z$. The histories we consider have a time step $\Delta t$ between the initial state (chosen to be $\rho = |+\rangle\langle+|$, with $|+\rangle = 1/\sqrt{2}(|0\rangle + |1\rangle)$) and first projector, as well as between the first and second projector, chosen so that $\gamma B\Delta t = 2\,\mathrm{rad}$. In addition, we only consider projectors onto the $xy$ plane of the Bloch sphere, parameterized by their azimuth. For this model, Fig. 3 shows the landscape of the cost in (9) for the ibmqx5 quantum processor[35] as well as a simulator whose precision was limited by imposing the same finite statistics as were collected with the quantum processor. Several minima found by running VCH on ibmqx5 are superimposed on the landscape (all points found below a noise threshold were considered to be equally valid minima). As these minima correspond reasonably well to theoretically consistent families, this represents a successful proof-of-principle implementation of VCH.

*Chiral molecule*. To highlight applications that will be possible on future hardware, we now turn to a simulated use of VCH to observe the quantum-to-classical transition for a chiral molecule[36,37]. The chiral molecule has been modeled as a two

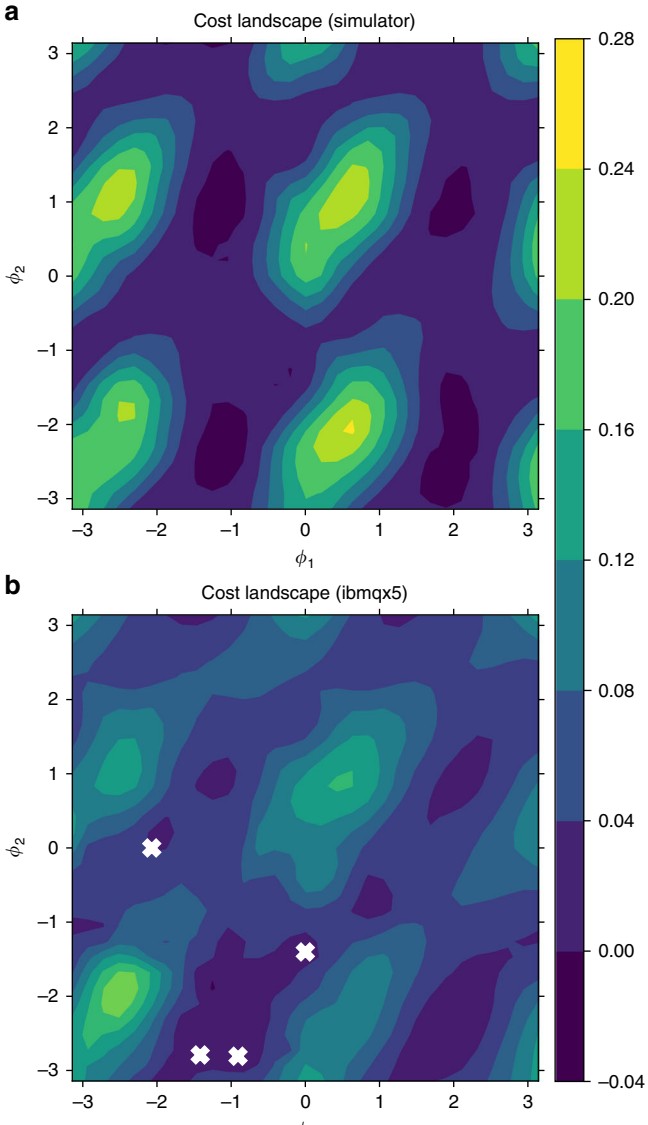

**Fig. 3** Consistency of three-time histories for a spin-1/2 particle in a magnetic field, with initial state $\rho = |+\rangle\langle+|$. The full-trace cost landscape, $C(\phi_1, \phi_2)$, is plotted as a function of the azimuths, $\phi_1$ and $\phi_2$, of the first and second projection bases, which we constrained to the $xy$ plane of the Bloch sphere. The point $(0, 0)$ corresponds to both projections being along the $x$ axis. Consistency is expected everywhere along certain vertical lines ($\phi_1 = 2 + n\pi\,\mathrm{rad}$), as they correspond to histories where the initial state is one of the projectors after the first time step, so there are no branches to interfere in the second time step. In addition, some slope-one lines ($\phi_2 = \phi_1 + (2 + n\pi)\mathrm{rad}$) should be consistent, as they correspond to histories where the second projectors are the same as the first after time evolution, so no interference occurs in the second time step. Indeed, one can see valleys in the cost landscapes for these vertical and slope-one lines, when the cost is quantified on a simulator **a** and on the ibmqx5 quantum computer **b**. It is noteworthy that negative cost values are possible due to finite statistics. The white "x" symbols in **b** mark some of the non-unique minima that the VCH algorithm found

level system where the right $|R\rangle$ and left $|L\rangle$ chirality states are described as $|R\rangle/|L\rangle = |+\rangle/|-\rangle = \frac{1}{\sqrt{2}}(|0\rangle \pm |1\rangle)$[37]. A chiral molecule in isolation would tunnel between $|R\rangle$ and $|L\rangle$, but we consider the molecule to be in a gas, where collisions with other molecules convey information about the molecule's chirality to its

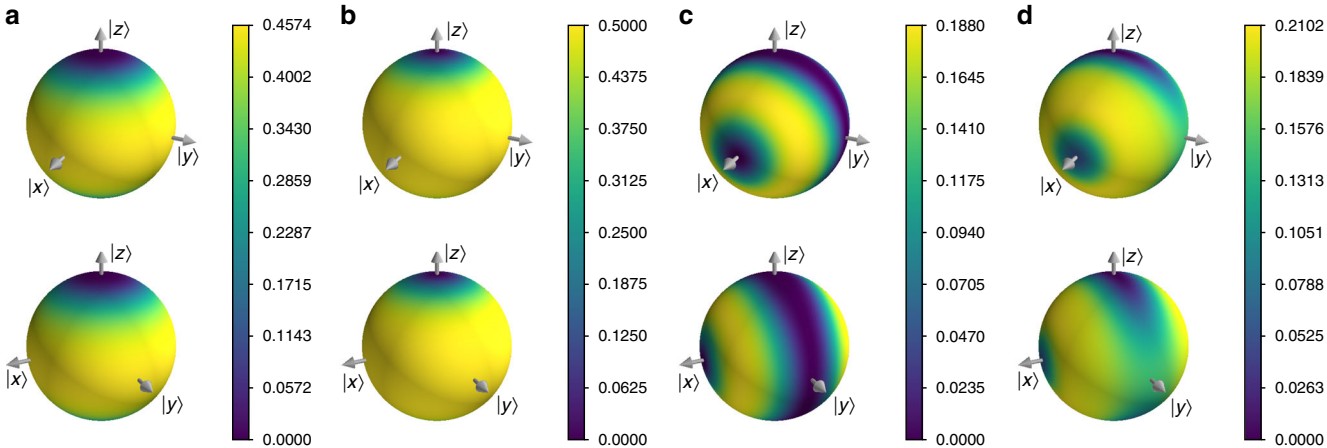

**Fig. 4** The cost landscape for stationary histories of the chiral molecule. As the projectors in these stationary histories are always along a single axis, we plot the cost on points where this axis would intersect the surface of the Bloch sphere. The bottom row of spheres are the same as the top, but rotated for additional perspective. Panels **a**, **b** show the full and partial-trace cost functions, respectively, for the case where the environment interactions are negligible ($\theta_z = 5$ rad, $\theta_x = 0.01$ rad) and thus we find that the energy eigenbasis ($z$ axis) is the only consistent stationary family as all others will branch as they evolve. In contrast, **c**, **d** are the full and partial-trace cost functions, respectively, for the case where the environment interactions dominate ($\theta_z = 0.01$ rad, $\theta_x = 5$ rad). One can see in **c**, **d** a significant difference between the full and partial-trace costs for the $y$ axis, meaning that this family of histories is consistent but not classical. In this regime, we also see that the chirality basis (the $x$ axis) is a local minimum for both cost functions and thus is approximately consistent and classical. For this chirality basis family, there is a ~0.01% chance that the molecule will change chirality during the evolution, showing that the quantum-to-classical transition leaves this system in a stabilized chiral state

environment. This information transfer is modeled by a rotation by angle $\theta_x$ about the $x$ axis of an environment qubit, controlled on the system's chirality, and for simplicity we suppose such collisions are evenly spaced at five points in time (see the Supplementary Note 5 for further details). We then consider simple families of stationary histories[37], where the projector set corresponds to the same basis at all five times (just after a collision occurs). Letting $\theta_z$ be the precession angle due to tunneling in the time between collisions, we can then explore the competition between decoherence and tunneling. Figure 4 shows our results for this model. Notably, we observe the transition from a quantum regime, where the chirality is not consistent, to a classical regime, where the chirality is both consistent and stable over time.

## Discussion

We expect VCH to revitalize interest in the CH approach to QM by increasing its practical utility. Making it possible to apply the tools and concepts of quantum foundations to a wide array of physical situations, as VCH will, is an important step for our understanding of the physical world. Specifically by providing an exponential speedup and reduction in resources over classical methods, VCH will provide a way to study phenomena including the quantum-to-classical transition[31,32,38], dynamics of quantum phase transitions[39], quantum biological processes[40], conformational changes[41], and many other complex phenomena that so far have been computationally intractable. In addition, VCH could be applied to study quantum algorithms themselves[42]. In order to highlight such potential applications and examine their resource requirements, we now focus on two of them: the emergence of classical diffusive dynamics in quantum spin systems and the appearance of defined pathways in protein folding.

In the context of nuclear magnetic resonance (NMR) experiments, it has long been known that systems with many spins obey a classical diffusion equation while smaller spin systems undergo Rabi oscillations. Despite the long history of spin diffusion studies[43–45], there is still no derivation of the transition from quantum oscillations to classical diffusion that can predict the size of the system where we should find that transition, or the

nature of the transition. Applying VCH to the study of histories of spin systems would clarify this point by showing the scale and abruptness with which the diffusive behavior emerges. As spin diffusion has been observed for systems as small as ~30,000 spins[46], we estimate that between ~$10^2$ and ~$10^3$ qubits would allow us to study this transition. For more details about this application, see the Supplementary Note 6.

In the protein-folding community there are currently two main schools of thought on how proteins fold. The first is that proteins fold along well-determined pathways with discrete folding units (foldons)[47], whereas the second is that there should be a funnel in the energy landscape of folding configurations, causing the system to explore a wide range of configurations before settling into the final state[48]. The deterministic pathways of the foldon model are favored by NMR experiments, raising the question of whether these views can be reconciled[47]. By providing the means to study the dynamic emergence of classical paths, i.e., the quantum-to-classical transition for proteins, VCH could resolve this discrepancy. For this purpose, we estimate that between ~$10^3$ and ~$10^4$ qubits will be needed. See the Supplementary Note 6 for more details on this application and resource estimate.

Finally, our work highlights the synergy of two distinct fields, quantum foundations and quantum computational algorithms, and hopefully will inspire further research into their intersection.

## Methods

**Evaluation of the cost**. Figure 5 shows the circuits for computing the full-trace cost (partial-trace cost) from two copies of $\sigma^A$ ($\sigma^{SA}$). It is noteworthy that both costs can be written as a difference of purities:

$$C = \mathrm{Tr}((\sigma^A)^2) - \mathrm{Tr}(\mathcal{Z}^A(\sigma^A)^2) \tag{11}$$

$$C_{pt} = \mathrm{Tr}((\sigma^{SA})^2) - \mathrm{Tr}(\mathcal{Z}^A(\sigma^{SA})^2). \tag{12}$$

The $\mathrm{Tr}((\sigma^A)^2)$ and $\mathrm{Tr}((\sigma^{SA})^2)$ terms are computed via the Swap Test, with a depth-two circuit and classical post-processing that scales linearly in the number of qubits[49,50]. A similar but even simpler circuit, called the Diagonalized Inner Product (DIP) Test[26], calculates the $\mathrm{Tr}(\mathcal{Z}^A(\sigma^A)^2)$ term with a depth-one circuit and no post-processing. Finally, the $\mathrm{Tr}(\mathcal{Z}^A(\sigma^{SA})^2)$ term is evaluated with the Partial-DIP Test[26], a depth-two circuit that is a hybridization of the Swap Test and the DIP Test.

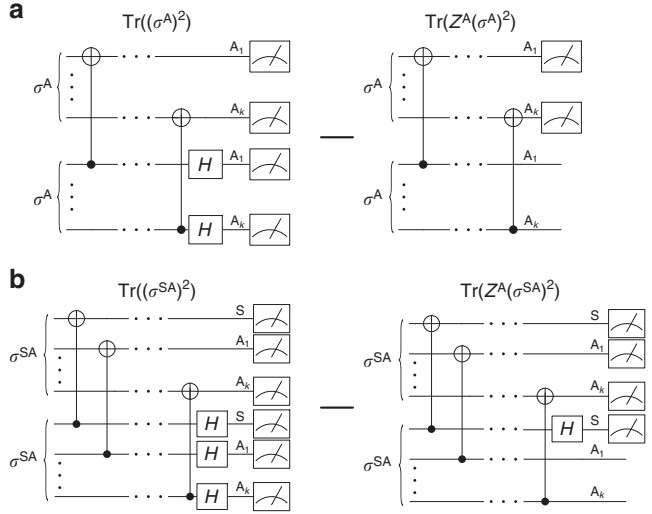

**Fig. 5** Circuits for computing the cost functions. Panel **a** shows the circuits for the full-trace cost $C$ function and **b** shows the circuit for the partial-trace cost $C_{pt}$

**Precision of probability readout.** One does not know a priori how many histories will be characterized in the probability readout step (Fig. 2c). Due to statistical noise, the probability of histories with greater probability will be determined with greater relative precision than those with lesser probability. Hence, it is reasonable to set a precision (or statistical noise) threshold, $\varepsilon$. Let $N_{readout}$ be the number of repetitions of the probability readout circuit. Then, histories $\mathcal{Y}^{\alpha}$ whose bitstring $\alpha$ occurs with frequency $f_{\alpha} < \sqrt{N_{readout}}/\varepsilon_{max}$ should be ignored, as their probabilities $p(\alpha) = f_{\alpha}/N_{readout}$ were not characterized with the desired precision. We separate $\mathcal{F}$ into the set $\mathcal{F}_c$ of histories whose probabilities are above the precision threshold (which we previously referred to loosely as the most likely histories) and the set of all other histories in $\mathcal{F}$:

$$\mathcal{F} = \mathcal{F}_c \cup \overline{\mathcal{F}_c}. \tag{13}$$

Computational complexity can be hidden in the value of $N_{readout}$ needed to obtain a desired precision for the probabilities of histories of interest. This issue is closely connected to the entropy of the set $\{\mathcal{D}(\alpha, \alpha)\}$, or equivalently, the entropy of the quantum state $\mathcal{Z}^A(\sigma^A)$. When $\mathcal{Z}^A(\sigma^A)$ is high entropy, an exponentially large number of histories may have non-zero probability and hence $N_{readout}$ would need to grow exponentially. VCH is therefore better suited to applications where there is a small subset of the histories that are far more probable than the rest. In the parameter optimization loop, one can select for families with this property by penalizing families for which $\mathcal{Z}^A(\sigma^A)$ has high entropy. Specifically, by noting that $P := \text{Tr}(\mathcal{Z}^A(\sigma^A)^2)$ can be efficiently computed via the circuit in Fig. 5a, one can modify the costs functions in Eqs (9) and (10) to be $\tilde{C} = C/P$ and $\tilde{C}_{pt} = C_{pt}/P$.

We remark that classicality is intimately connected to predictability, with the emergence of classicality linked to the so-called predictability sieve[51,52]. As the CH formalism is typically used to find classical families, this implies predictable families (i.e., families with low entropy or high purity $P$) are arguably of the most interest. Hence, our modified cost function $\tilde{C}$ also serves to select those consistent families with histories that are the most predictable and therefore the most classical.

**Approximate consistency.** Here we discuss how VCH outputs an upper bound on the consistency parameter $\varepsilon$. Let us first relate the cost $C$ to $\varepsilon$. For any pair of histories $\mathcal{Y}^{\alpha}$ and $\mathcal{Y}^{\alpha'}$ in $\mathcal{F}$,

$$|\mathcal{D}(\alpha, \alpha')|^2 \le C/2, \tag{14}$$

which follows from Eq. (9) and the fact that $|\mathcal{D}(\alpha, \alpha')| = |\mathcal{D}(\alpha', \alpha)|$. Let us define

$$\varepsilon_{\alpha, \alpha'} := \sqrt{\frac{C}{2\mathcal{D}(\alpha, \alpha)\mathcal{D}(\alpha', \alpha')}}. \tag{15}$$

Then it follows from Eq. (14) that

$$|\mathcal{D}(\alpha, \alpha')|^2 \le \varepsilon_{\alpha, \alpha'}^2 \mathcal{D}(\alpha, \alpha)\mathcal{D}(\alpha', \alpha'), \tag{16}$$

which corresponds to the approximate consistency condition from Eq. (6). Hence, probablity sum rules for these two histories are satisfied within error $\varepsilon_{\alpha, \alpha'}$, which can be calculated from Eq. (15) for histories in $\mathcal{F}_c$, as the probabilites are known for these histories.

Next, consider histories in $\overline{\mathcal{F}_c}$. As we do not have enough information to differentiate these histories, we advocate combining the elements of $\overline{\mathcal{F}_c}$ into a single coarse-grained history $\mathcal{Y}^{\gamma}$.

Let $\mathcal{Y}^{\beta}$ be the least likely history in $\mathcal{F}_c$. Then defining $\delta^2 = \mathcal{D}(\gamma, \gamma)/\mathcal{D}(\beta, \beta)$, we can make use of the positive semi-definite property of $\sigma^A$ to write:

$$|\mathcal{D}(\gamma, \beta)|^2 \le \mathcal{D}(\gamma, \gamma)\mathcal{D}(\beta, \beta) = \delta^2 \mathcal{D}(\beta, \beta)^2. \tag{17}$$

As $\mathcal{Y}^{\beta}$ is the least likely history in $\mathcal{F}_c$, this expression then lets us bound the error on the probability sum rule (giving a weaker approximate consistency condition[30]) between $\mathcal{Y}^{\gamma}$ and any $\mathcal{Y}^{\alpha} \in \mathcal{F}_c$ as:

$$\begin{aligned} |\mathcal{D}(\gamma, \alpha)| &\le \delta\mathcal{D}(\alpha, \alpha) \\ &\le \delta(\mathcal{D}(\gamma, \gamma) + \mathcal{D}(\alpha, \alpha)). \end{aligned} \tag{18}$$

It is then possible to characterize the approximate consistency of the histories of $\mathcal{F}$ pairwise with $\varepsilon_{\alpha, \alpha'}$ and $\delta$. Alternatively, to give an upper bound on the overall consistency $\varepsilon$, we take the greatest of these pairwise bounds:

$$\varepsilon \le \max(\{\varepsilon_{\alpha, \alpha'}\} \cup \{\delta\}). \tag{19}$$

For those applications where we are working with the partial-trace consistency, the notion of approximate consistency is somewhat more obscured. In order to generate probabilities and bound $\varepsilon$, we therefore recommend evaluating the full-trace cost function at the minimum found with the partial-trace cost. This approach is helpful, as any partial-trace consistent family will also be full-trace consistent and the partial-trace consistency does not directly allow one to discuss probabilities in the same way. Taking this approach allows us to then directly utilize the approximate consistency framework above.

## Data Availability
The data used to create the figures in this article are available upon request. Requests should be sent to the corresponding author.

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

## Acknowledgements

We thank IBM for the use of their quantum processor. The views expressed in this article are those of the authors and not of IBM. This work was supported by the U.S. Department of Energy (DOE), Office of Science, Office of High Energy Physics, QuantISED program, and also by the U.S. DOE, Office of Science, Basic Energy Sciences, Materials Sciences and Engineering Division, Condensed Matter Theory Program. All authors acknowledge support from the LDRD program at Los Alamos National Laboratory (LANL). L.C. was also supported by the DOE through the J. Robert Oppenheimer fellowship. A.T.S. and P.J.C. additionally acknowledge support from the LANL ASC Beyond Moore's Law project. Finally, W.H.Z. acknowledges partial support by the Foundational Questions Institute grant FQXi-1821 and Franklin Fetzer Fund, a donor advised fund of the Silicon Valley Community Foundation.

## Author contributions

All authors contributed to the preparation and revision of the manuscript. P.J.C. invented the algorithm and developed the basic formalism. A.A. designed and carried out the experimental implementations, analyzed the results, and contributed to the formalism. L. C., A.T.S., and W.H.Z. consulted on all stages of the project.

## Additional information

**Competing interests:** The authors declare no competing interests.

