## [Peer Review File · Nature Communications]

Reviewers' comments:

Reviewer #1 (Remarks to the Author):

The central theme in the paper by Arrasmith et al. are the so-called consistent histories. This concept goes back to the 80's where it has been suggested in order to make sense of the fact that we experience almost exclusively classical behavior while (most likely) living in a quantum world. The consistent history approach suggests, that quantum and classical behavior may be in fact nearly indistinguishable, depending on the system and the measurements we are allowed to perform. (Consistent or decoherent histories are, loosely speaking, the opposite of what we have in a double slit experiment) . A major drawback of the consistency approach has always been the computational inaccessibility: Given even a moderately complex quantum system and a reasonable set of measurements, it is computationally extremely hard to decide whether or not the corresponding histories are indeed consistent, i.e., „quasiclassical“.

Overcoming this drawback is the aim of the present paper: Arrasmith et al. present an algorithm to efficiently evaluate the consistency of some complex set-up. It furthermore allows the optimization of consistency with respect to measurements. The crucial routine that evaluates the consistency is a quantum algorithm. If this algorithm could indeed be run on a quantum computer, this would most likely cause a leap forward in the consistency based research on the foundation of quantum physics.

Overall the paper is clearly written and the algorithm is to my best knowledge original. The result is timely and possibly of major impact. In principle I would thus recommend publication of the paper. There are however a few points that I would ask the authors to address prior to a final decision:

1. At the end of the first paragraph it reads: „...the emergence of classical behavior (objectivity, irreversibility, lack of interference, etc.)...“ Should „irreversibility“ really be in this list? At the bottom quantum and classical physics are reversible both alike.
2. Near the end of the introduction it reads: „Hence, useful implementations of our algorithm will be feasible on near-term quantum devices.“ This may be perceived a bit optimistic (speculative).
3. The Hamiltonian of the first example reads $H = -\gamma B \sigma_z$. What is the concrete value of γB that was used to compute Fig. 3 ? What is the time elapsed before and between the two measurements?
4. In the caption of Fig. 3 the initial state is $|+\rangle\langle+|$. What is this state? Is its definition similar to the one given later in the context of the chiral molecule?
5. What is the „simulator“ (Fig. 3a) ? The first example is simple enough to allow for a sufficiently precise computation of the cost landscape using standard numerical means. So is the landscape from the simulator practically exact?
6. Also in the caption of Fig. 3 it reads: „Note that negative cost values are possible due to finite statistics“. What statistics does that refer to? I guess the simulator does not need to rely on any statistics?
7. What are the „non-unique“ minima marked by the white x-signs in Fig. 3 b? What is special about them? Are they also found by the simulator? And if not, why?
8. The Section on „experimental realizations“ is a bit poorly structured. The subsections are „quantum hardware“ and „simulator“ but indeed two different examples are discussed in the two subsections. The clarity of presentation of this Section could be improved.

Jochen Gemmer

Reviewer #2 (Remarks to the Author):

The paper by Arrasmith and collaborators discusses the use of a quantum computer within the consistent histories framework to address quantum foundational problems. Although the algorithm presented uses many of the same tools these authors have used in previous studies, the work describes a new way to use variational algorithms to answer interesting questions, and shows a proof of principle computation on an experimental device. While this is in-principle a novel way to use a quantum device, what is less clear to me in terms of impact that could result from this use. While the work has this potential, I think the authors could strengthen the exposition in a few ways to better merit publication, which I detail below.

In particular, many works on quantum foundations posit the power of the approach to answer fundamental questions about interpretation and connection to classical correspondence, but often do so in an abstract sense. To better gauge the potential for impact, it would be helpful if the authors could posit at least one use case they would apply the technique to in the aspirational limit of many/perfect qubits to get a result that has not yet been classically accessible. It does not need to be performed on a device, but rather a sketch of the question to be answered, which system it would be applied to, how one uses their technique to measure it, and about how much resources one might need. It is difficult for someone outside of quantum foundations to know what measurements or results would settle an unanswered question. If no concrete use cases are known where one wants to measure a beyond classical result that would answer a real foundational question, it may be safe to assume this work will minimal impact.

On the more technical side, I find one aspect a bit troubling that I hope the authors can correct me on. In particular, there are likely to be an almost infinite number (in a continuous construction) of possible families of consistent histories. This could lead to a large redundancy in the optimization landscape, where even at convergence one can drift between different consistent families. The assertion that sampling is efficient and one finds a dominant set of classical paths would seem to depend on having found a family of histories where such classical paths are ingrained in the projectors. If I randomize these projectors continuously, but still get a consistent set of histories, I will likely find an inefficient distribution, circumventing most of the useful aspects of this approach, and rendering it somewhat useless. In the body and text of the methods section, the authors assert that one can penalize such high entropy consistent histories through a particular measurement. However, it seems like this could lead to projectors which have quantum character (project into superpositions of spins for example) in order to minimize this entropy measure, which seems like it defeats the purpose of a consistent histories approach telling the story in a classically understandable way. I imagine in the non-variational version of consistent histories, these projectors are probably just chosen this way. I feel to strengthen the paper, the authors need to go into more detail as to how their approach can find paths that are both **likely** and **classically interpretable**, otherwise I can't really see the utility in the approach.

Reviewer #3 (Remarks to the Author):

This is an interesting piece of work which covers a wide variety of aspects and applications of the field of decoherent histories. I believe that with small changes it is suitable for publication and will, as the authors state, revitalize the field. I have only minor comments which I ask the authors to address.

1. There is only one small error that I could spot. The authors state after Eq.(4) that $D(a,a)$ is a probability if and only if E.(4) holds. Actually, we only need the real part of the LHS of Eq.(4). This

needs to be corrected. (And any subsequent consequences worked out). Alternatively, the authors could observe that although only the real part is required but, typically, both real and imaginary parts vanish hence it is no particular loss to work with the stronger condition Eq.(4).

2. The authors make the general remark early on about how the number of histories increases dramatically with system size and number of times, and hence the computational intractability increases correspondingly. I make one small comment here which is that for linear systems (and perturbation about them) described by continuous variables one can often solve for the decoherence functional in essentially exact terms (even though one has an infinite dimensional Hilbert space). Some brief (optional) discussion around this case may be useful by way of contrast to the finite dimensional systems considered here.

3. A closely related situation to consider is the linear positive condition of Goldstein and Page, in which the histories are assigned a quasi-probability $q(a) = \text{Re Tr} (C^a \rho)$. When positive it is a probability satisfying all sum rules exactly and is an alternative but weaker condition to the decoherence condition Eq.(4) (and also involves a smaller number of component conditions). These conditions have been the focus of some interest over the years so it may be a nice extension of the present work to show how to apply the technique to this case. Also, note that $q(a) < 0$ implies that the histories must be inconsistent so this is a potentially cheaper way of identifying inconsistent histories which does not involve all the off-diagonal terms of Eq.(3). This is again an optional change.

Reviewer #1 (Remarks to the Author):

The central theme in the paper by Arrasmith et al. are the so-called consistent histories. This concept goes back to the 80's where it has been suggested in order to make sense of the fact that we experience almost exclusively classical behavior while (most likely) living in a quantum world. The consistent history approach suggests, that quantum and classical behavior may be in fact nearly indistinguishable, depending on the system and the measurements we are allowed to perform. (Consistent or decoherent histories are, loosely speaking, the opposite of what we have in a double slit experiment) . A major drawback of the consistency approach has always been the computational inaccessibility: Given even a moderately complex quantum system and a reasonable set of measurements, it is computationally extremely hard to decide whether or not the corresponding histories are indeed consistent, i.e., „quasiclassical“.

Overcoming this drawback is the aim of the present paper: Arrasmith et al. present an algorithm to efficiently evaluate the consistency of some complex set-up. It furthermore allows the optimization of consistency with respect to measurements. The crucial routine that evaluates the consistency is a quantum algorithm. If this algorithm could indeed be

run on a quantum computer, this would most likely cause a leap forward in the consistency based research on the foundation of quantum physics.

Overall the paper is clearly written and the algorithm is to my best knowledge original. The result is timely and possibly of major impact. In principle I would thus recommend publication of the paper. There are however a few points that I would ask the authors to address prior to a final decision:

1. At the end of the first paragraph it reads: „...the emergence of classical behavior (objectivity, irreversibility, lack of interference, etc.)....“ Should „irreversibility“ really be in this list? At the bottom quantum and classical physics are reversible both alike.

In this context, we mean irreversibility in the sense of statistical mechanics associated with increasing entropy. This sense of irreversibility is important for the behavior of many aspects of the classical world we experience.

2. Near the end of the introduction it reads: „Hence, useful implementations of our algorithm will be feasible on near-term quantum devices.“ This may be perceived a bit optimistic (speculative).

We acknowledge that this is speculative and have changed “will” to be “may”. We note however that this speculation is not wildly unreasonable based on the recent progress that has been made with quantum computers.

3. The Hamiltonian of the first example reads $H = -\gamma B \sigma_z$. What is the concrete value of γB that was used to compute Fig. 3? What is the time elapsed before and between the two measurements?

Rather than specifying a specific value of γB , and Δt , we instead specify only their product (since that is all that matters for the current situation). For our purposes, we set $\gamma B \Delta t = 2$ radians. We have clarified this in the section “Spin in a Magnetic Field”.

4. In the caption of Fig. 3 the initial state is $|+\rangle\langle+|$. What is this state? Is its definition similar to the one given later in the context of the chiral molecule?

We have also clarified this point in the section “Spin in a Magnetic Field”.

$$|+\rangle = 1/\sqrt{2} (|0\rangle + |1\rangle)$$

5. What is the „simulator“ (Fig. 3a) ? The first example is simple enough to allow for a sufficiently precise computation of the cost landscape using standard numerical means. So is the landscape from the simulator practically exact?

Our numerical simulator is fairly standard and capable of being essentially exact, but we choose to reproduce the “noise” of the finite statistics associated with the number of samples that we took on the physical quantum computer we used. This is now clarified in the text.

6. Also in the caption of Fig. 3 it reads: „Note that negative cost values are possible due to finite statistics“. What statistics does that refer to? I guess the simulator does not need to rely on any statistics?

The statistics are those mentioned in response to point 5.

7. What are the „non-unique“ minima marked by the white x-signs in Fig. 3 b? What is special about them? Are they also found by the simulator? And if not, why?

In principle, there is nothing special about these points. They just happen to have been found on the noisy quantum computer by imposing a noise cutoff on points found with a minimization procedure. This has also been clarified in the text.

8. The Section on „experimental realizations“ is a bit poorly structured. The subsections are „quantum hardware“ and „simulator“ but indeed two different examples are discussed in the two subsections. The clarity of presentation of this Section could be improved.

We agree that the presentation here needed improvement and have now changed the subsection headings to hopefully improve the clarity.

Jochen Gemmer

Reviewer #2 (Remarks to the Author):

The paper by Arrasmith and collaborators discusses the use of a quantum computer

within the consistent histories framework to address quantum foundational problems. Although the algorithm presented uses many of the same tools these authors have used in previous studies, the work describes a new way to use variational algorithms to answer interesting questions, and shows a proof of principle computation on an experimental device. While this is in-principle a novel way to use a quantum device, what is less clear to me in terms of impact that could result from this use. While the work has this potential, I think the authors could strengthen the exposition in a few ways to better merit publication, which I detail below.

In particular, many works on quantum foundations posit the power of the approach to answer fundamental questions about interpretation and connection to classical correspondence, but often do so in an abstract sense. To better gauge the potential for impact, it would be helpful if the authors could posit at least one use case they would apply the technique to in the aspirational limit of many/perfect qubits to get a result that has not yet been classically accessible. It does not need to be performed on a device, but rather a sketch of the question to be answered, which system it would be applied to, how one uses their technique to measure it, and about how much resources one might need. It is difficult for someone outside of quantum foundations to know what measurements or results would settle an unanswered question. If no concrete use cases are known where one wants to measure a beyond classical result that would answer a real foundational question, it may be safe to assume this work will minimal impact.

In response to this point, we have added additional discussion on two concrete use cases in the “Discussion” section, and expanded on them in the Supplementary Material. In particular, we focused on two systems for which experimentalists have observed a sort of quantum-to-classical transition, but for which there is no theoretical treatments of these transitions. The first example is **spin diffusion**, where the scale and abruptness of the transition is unknown. Exploring this may be relevant to the development of spin qubits for quantum computing. The second example is **protein folding**, where there is an ongoing debate about whether proteins fold by many pathways or by a single deterministic pathway. Applying VCH to this case could help to resolve this debate. Please see our revised Discussion section, as well as our newly added Appendix F, for details about these applications, including estimates for the qubit resource requirements.

On the more technical side, I find one aspect a bit troubling that I hope the authors can correct me on. In particular, there are likely to be an almost infinite number (in a continuous construction) of possible families of consistent histories. This could lead to a large redundancy in the optimization landscape, where even at convergence one can drift

between different consistent families. The assertion that sampling is efficient and one finds a dominant set of classical paths would seem to depend on having found a family of histories where such classical paths are ingrained in the projectors. If I randomize these projectors continuously, but still get a consistent set of histories, I will likely find an inefficient distribution, circumventing most of the useful aspects of this approach, and rendering it somewhat useless. In the body and text of the methods section, the authors assert that one can penalize such high entropy consistent histories through a particular measurement. However, it seems like this could lead to projectors which have quantum character (project into superpositions of spins for example) in order to minimize this entropy measure, which seems like it defeats the purpose of a consistent histories approach telling the story in a classically understandable way. I imagine in the non-variational version of consistent histories, these projectors are probably just chosen this way. I feel to strengthen the paper, the authors need to go into more detail as to how their approach can find paths that are both *likely* and *classically interpretable*, otherwise I can't really see the utility in the approach.

We understand the concern here. However, we wish to reassure the reviewer that there is a very strong connection between pathways that are “likely” and those that are “classically interpretable”. There is an extensive body of literature (e.g., see the newly added references [51,52]) that connects classicality to predictability. This concept (or phenomenon) is called the predictability sieve, and it was partially developed by one of the authors (Zurek). The idea is that families of histories that we associate with classical dynamics also happen to correspond to families that have low entropy (high predictability) relative to other families for the same system.

In the context of our VCH algorithm, the situation is fortuitous. We are lucky that the families for which VCH may be efficient (low entropy families) also happen to correspond to the families in which we are most interested (classical families).

The benefit of our proposed alternative cost function \tilde{C} is that we are actually biasing our search towards histories that are more predictable and hence more classical (via Refs [51,52]). We have added a mention of this connection at the end of the “Precision of probability readout” sub-section in the “Methods”.

Reviewer #3 (Remarks to the Author):

This is an interesting piece of work which covers a wide variety of aspects and applications of the field of decoherent histories. I believe that with small changes it is

suitable for publication and will, as the authors state, revitalize the field. I have only minor comments which I ask the authors to address.

1. There is only one small error that I could spot. The authors state after Eq.(4) that $D(a,a)$ is a probability if and only if E.(4) holds. Actually, we only need the real part of the LHS of Eq.(4). This needs to be corrected. (And any subsequent consequences worked out). Alternatively, the authors could observe that although only the real part is required but, typically, both real and imaginary parts vanish hence it is no particular loss to work with the stronger condition Eq.(4).

We thank the reviewer for pointing this out. Indeed, we had left out a mention that the weak decoherence condition (where the only the real part is required to vanish) suffices for a lack of interference. We have clarified this point in the text.

2. The authors make the general remark early on about how the number of histories increases dramatically with system size and number of times, and hence the computational intractability increases correspondingly. I make one small comment here which is that for linear systems (and perturbation about them) described by continuous variables one can often solve for the decoherence functional in essentially exact terms (even though one has an infinite dimensional Hilbert space). Some brief (optional) discussion around this case may be useful by way of contrast to the finite dimensional systems considered here.

We have added a brief comment in the introduction acknowledging that there are also some cases of simple continuous systems where path integral approaches can facilitate the evaluation of the decoherence functional, along with a relevant citation.

3. A closely related situation to consider is the linear positive condition of Goldstein and Page, in which the histories are assigned a quasi-probability $q(a) = \text{Re Tr} (C^a \rho)$. When positive it is a probability satisfying all sum rules exactly and is an alternative but weaker condition to the decoherence condition Eq.(4) (and also involves a smaller number of component conditions). These conditions have been the focus of some interest over the years so it may be a nice extension of the present work to show how to apply the technique to this case. Also, note that $q(a) < 0$ implies that the histories must be inconsistent so this is a potentially cheaper way of identifying inconsistent histories which does not involve all the off-diagonal terms of Eq.(3). This is again an optional change.

We do not feel that it would be helpful to include a discussion of this alternative as this work is about the consistent history framework. Additionally, while we can see a way to use methods similar to what we have used here for evaluating the linear positive condition for a single particular history, we can see no way to do so that is more efficient than evaluating the VCH cost function and thus getting all of the off-diagonal terms. We therefore also do not expect that there would be a computational advantage to the linear positive condition over consistent histories on quantum computers to motivate such an extension.

Sincerely,

Patrick Coles, LANL, pcoles@lanl.gov (Corresponding Author)

Copy: Andrew Arrasmith, LANL, aarrasmith@lanl.gov

Lukasz Cincio, LANL, lcincio@lanl.gov

Andrew Sornborger, LANL, sornborg@lanl.gov

Wojciech Zurek, LANL, whz@lanl.gov

Variational Consistent Histories: A Hybrid Algorithm for Quantum Foundations

Andrew Arrasmith,^{1,2} Lukasz Cincio,¹ Andrew T. Sornborger,³ Wojciech H. Zurek,¹ and Patrick J. Coles¹

¹*Theoretical Division, Los Alamos National Laboratory, Los Alamos, NM 87545, USA.*

²*Department of Physics, University of California Davis, Davis, CA 95616, USA.*

³*Information Sciences, Los Alamos National Laboratory, Los Alamos, NM 87545, USA.*

While quantum computers are predicted to have many commercial applications, less attention has been given to their potential for resolving foundational issues in quantum mechanics. Here we focus on quantum computers' utility for the Consistent Histories formalism, which has previously been employed to study quantum cosmology, quantum paradoxes, and the quantum-to-classical transition. We present a variational hybrid quantum-classical algorithm for finding consistent histories, which should revitalize interest in this formalism by allowing classically impossible calculations to be performed. In our algorithm, the quantum computer evaluates the decoherence functional (with exponential speedup in both the number of qubits and the number of times in the history), and a classical optimizer adjusts the history parameters to improve consistency. We implement our algorithm on a cloud quantum computer to find consistent histories for a spin in a magnetic field, and on a simulator to observe the emergence of classicality for a chiral molecule.

The foundations of quantum mechanics (QM) have been debated for the past century [1, 2], including topics such as the EPR paradox, hidden-variable theories, Bell's Theorem, Born's rule, and the role of measurements in QM. This also includes the quantum-to-classical transition, i.e., the emergence of classical behavior (objectivity, irreversibility, lack of interference, etc.) from quantum laws [3–5].

The Consistent Histories (CH) formalism was introduced by Griffiths, Omnès, Gell-Mann, and Hartle to address some (though not all) of the aforementioned issues [6–8]. One inventor considered CH to be “the Copenhagen interpretation done right” [6], as it resolves some of the paradoxes of quantum mechanics by enforcing strict rules for logical reasoning with quantum systems. In this formalism, the Copenhagen interpretation's focus on measurements as the origin of probabilities is replaced by probabilities for sequences of events (histories) to occur, and hence by avoiding measurements it avoids the measurement problem. The sets of histories whose probabilities are additive (as the histories do not interfere with each other) are considered to be consistent and are thus the only ones able to be reasoned about in terms of classical probability and logic [7].

Regardless of one's opinion of the philosophical interpretation (on which this paper is agnostic), this computational framework has proven useful in applications such as attempting to solve the cosmological measure problem [9, 10], understanding quantum jumps [11], and evaluating the arrival time for particles at a detector [12–14]. One of the main reasons that this framework has not received more attention and use is that carrying out the calculations for non-trivial cases (e.g., discrete systems of appreciable size or continuous systems that do not admit approximate descriptions by exactly solvable path integrals) can be difficult [11][11, 15]. While numerical approaches have been attempted [16, 17], they require exponentially scaling resources as either the number of times considered or the system size grows. This makes classical numerical approaches unusable for any but the

simplest cases.

Here we present a scalable algorithm for the CH formalism that achieves an exponential speedup over classical methods both in terms of the system size and the number of times considered. It will allow exploration beyond toy models, such as the quantum-to-classical transition in mesoscopic quantum systems. We expect this to revitalize interest in the CH approach to quantum mechanics by increasing its practical utility.

Our algorithm is a variational hybrid quantum-classical algorithm (VHQA). With the impending arrival of the first noisy intermediate-scale quantum computers [18], the field of VHQAs, which make the most of short quantum circuits combined with classical optimizers, has been taking off. VHQAs have now been demonstrated for a myriad of tasks ranging from factoring to finding ground states, among others [19–26]. The VHQA framework potentially brings the practical applications of quantum computers years closer to fruition. Hence, useful implementations of our algorithm ~~will~~ may be feasible on near-term quantum devices.

Below we introduce our algorithm, present experimental results from implementing our method on IBM's superconducting qubit quantum processor as well as on a simulator, and then discuss future applications.

CONSISTENT HISTORIES BACKGROUND

In the CH framework [27–29], a history \mathcal{Y}^α is a sequence of properties (i.e., projectors onto the appropriate subspaces) at a succession of times $t_1 < t_2 < \dots < t_k$,

$$\mathcal{Y}^\alpha = (P_1^{\alpha_1}, P_2^{\alpha_2}, \dots, P_k^{\alpha_k}), \quad (1)$$

where $P_j^{\alpha_j}$ is chosen from a set P_j of projectors that sum to the identity at time t_j . For example, for a photon passing through a sequence of diffraction gratings and then striking a screen, a history could be the photon passed through one slit in the first grating, another slit in the

second, and so on. Clearly, we find interference between such histories unless there is some sense in which the photon's path has been recorded. Since there is interference, we cannot add the probabilities of the different histories classically and expect to correctly predict where the photon strikes the screen.

The CH framework provides tools for determining when a family (i.e., a set that sums to the multi-time identity operator) of histories $\mathcal{F} = \{\mathcal{Y}^\alpha\}$ exhibits interference, which is not always obvious. In this framework, one defines the so-called class operator

$$\mathcal{C}^\alpha = P_k^{\alpha_k}(t_k)P_{k-1}^{\alpha_{k-1}}(t_{k-1})\dots P_1^{\alpha_1}(t_1), \quad (2)$$

which is the time-ordered product of the projection operators (now in the Heisenberg picture and hence explicitly time dependent) in history \mathcal{Y}^α . If the system is initially described by a density matrix ρ , the degree of interference or overlap between histories \mathcal{Y}^α and $\mathcal{Y}^{\alpha'}$ is

$$\mathcal{D}(\alpha, \alpha') = \text{Tr}(\mathcal{C}^\alpha \rho \mathcal{C}^{\alpha'\dagger}). \quad (3)$$

This quantity is called the decoherence functional. The consistency condition for a family of histories \mathcal{F} is then

$$\text{Re}(\mathcal{D}(\alpha, \alpha')) = 0, \quad \forall \alpha \neq \alpha'. \quad (4)$$

If and only if this condition holds do we say that $\mathcal{D}(\alpha, \alpha')$ is the probability for history \mathcal{Y}^α . For computational convenience, we will instead work with a stronger condition [28]:

$$\mathcal{D}(\alpha, \alpha') = 0, \quad \forall \alpha \neq \alpha', \quad (5)$$

Since we are presenting a numerical algorithm, it will also be useful to consider approximate consistency, where we merely insist that the interference is small in the following sense:

$$|\mathcal{D}(\alpha, \alpha')|^2 \leq \epsilon^2 \mathcal{D}(\alpha, \alpha) \mathcal{D}(\alpha', \alpha'), \quad \forall \alpha \neq \alpha', \quad (6)$$

which guarantees that probability sum rules for \mathcal{F} are satisfied within an error of ϵ [30].

To study consistency arising purely from decoherence (i.e., records in the environment), researchers have proposed a functional that instead takes a partial trace over E , which is (a subsystem of) the environment [31, 32]:

$$\mathcal{D}_{\text{pt}}(\alpha, \alpha') = \text{Tr}_E(\mathcal{C}^\alpha \rho \mathcal{C}^{\alpha'\dagger}). \quad (7)$$

With this modification, the consistency condition is

$$\mathcal{D}_{\text{pt}}(\alpha, \alpha') = \mathbf{0}, \quad \forall \alpha \neq \alpha', \quad (8)$$

where $\mathbf{0}$ is the zero matrix. Instead of only signifying the lack of interference, partial-trace consistency singles out whether or not the records of the histories in the environment interfere. Note that the full-trace condition

FIG. 1. An illustration of the branching of histories for k time steps. A one-spin ($n = 1$) and two-spin ($n = 2$) system, respectively shown in panels **a** and **b**, have 2^k and 2^{2k} different histories. Here, $k = 3$ in **a** and $k = 2$ in **b**.

of Eq. (5) is satisfied when this partial-trace consistency is satisfied, but the converse does not hold [31].

With this formalism in hand, we can now see why classical numerical schemes for CH have faced difficulty. For example, consider histories of a collection of n spin-1/2 particles for k time steps, depicted in Fig. 1. The number of histories is 2^{nk} , and hence there are $\sim 2^{2nk}$ decoherence functional elements. Furthermore, evaluating each decoherence functional element $\mathcal{D}(\alpha, \alpha')$ requires the equivalent of a Hamiltonian simulation of the system, i.e., the multiplication of $2^n \times 2^n$ matrices. This means modern clusters would take centuries to evaluate the consistency of a family of histories with $k = 2$ time steps and $n = 10$ spins. Given this limitation, we can see why, for the most part, only toy models have been analyzed in this framework thus far.

HYBRID ALGORITHM FOR FINDING CONSISTENT HISTORIES

We refer to our VHQCA as Variational Consistent Histories (VCH), see Fig. 2. VCH takes as its inputs a physical model (i.e., an initial state ρ and a Hamiltonian H) and some ansatz for the types of projectors to consider. It outputs: (1) a family \mathcal{F} of histories that is (approximately) full and/or partial trace consistent in the form of projection operators prepared on a quantum computer, (2) the probabilities of the most likely histories \mathcal{Y}^α in \mathcal{F} , and (3) a bound on the consistency parameter ϵ .

VCH involves a parameter optimization loop, where a

FIG. 2. Flowchart for VCH. The goal of VCH is to take a physical model (panel **a**) and output an approximately consistent family \mathcal{F} of histories (**e**), their associated probabilities $\{p(\alpha)\}$ (**c**), and a measure ϵ of how consistent \mathcal{F} is (**d**). This is accomplished via a parameter optimization loop (**b**), which is a hybrid quantum-classical computation. Here the classical computer adjusts the projector parameters (contained in the gates $\{B_j(\theta)\}$, where $B_j(\theta)$ diagonalizes the P_j projectors) and a quantum computer returns the cost. Note that P_j denotes the set of Schrodinger-picture projectors at the j^{th} time. The optimal parameters are then used to compute the probabilities of the most likely histories in \mathcal{F} (panel **c**) and to prepare the projectors for any history in \mathcal{F} (panel **e**), where X is the Pauli- X operator). While the quantum circuits are depicted for a one-qubit system, the SM discusses the generalizations to multi-qubit systems, non-trivial environment E , coarse-grained histories, and branch-dependent histories.

quantum computer evaluates a cost function that quantifies the family’s inconsistency, while a classical optimizer adjusts the family (i.e., varies the projector parameters) to reduce the cost. Classical optimizers for VHQCAs are actively being investigated [26, 33], and one is free to

choose the classical optimizer on an empirical basis.

To compute the cost, note that the elements of the decoherence functional form a positive semi-definite matrix with trace one. In VCH, we exploit this property to encode \mathcal{D} in a quantum state σ^A , whose matrix elements are $\langle \alpha | \sigma^A | \alpha' \rangle = \mathcal{D}(\alpha, \alpha')$. Step **b** of Fig. 2 shows a quantum circuit that prepares σ^A . This circuit transforms an initial state $\rho \otimes |\mathbf{0}\rangle\langle\mathbf{0}|$ on systems SA , where S simulates the physical system of interest and A is an ancilla system, into a state σ^{SA} whose marginal is σ^A . For the full trace consistency, we introduce a global measure of the (in)consistency that quantifies how far σ^A is from being diagonal, which serves as our cost function:

$$C := \sum_{\alpha \neq \alpha'} |\mathcal{D}(\alpha, \alpha')|^2 = D_{\text{HS}}(\sigma^A, \mathcal{Z}^A(\sigma^A)), \quad (9)$$

where D_{HS} is the Hilbert-Schmidt distance and $\mathcal{Z}^A(\sigma^A)$ is the dephased (all off-diagonal elements set to zero) version of σ^A . This quantity goes to zero if and only if \mathcal{F} is consistent. For the partial trace case, we arrive at a similar cost function but with σ^A replaced by σ^{SA} :

$$C_{\text{pt}} := \sum_{\alpha \neq \alpha'} \|\mathcal{D}_{\text{pt}}(\alpha, \alpha')\|_{\text{HS}}^2 = D_{\text{HS}}(\sigma^{SA}, \mathcal{Z}^A(\sigma^{SA})). \quad (10)$$

Here the notation $\mathcal{Z}^A(\sigma^{SA})$ indicates that the dephasing operation only acts on system A , and the absolute squares of Eq. (9) have been generalized to Hilbert-Schmidt norms, $\|M\|_{\text{HS}}^2 := \text{Tr}(M^\dagger M)$. In the Methods section, we present quantum circuits that compute these cost functions from two copies of σ^A or σ^{SA} . Derivations of the second equalities in Eq. (9) and Eq. (10) can be found in the Supplementary Material (SM). We remark that alternative cost functions may be useful, for example, to penalize families \mathcal{F} with high entropy (see Methods) or to obtain a larger cost gradient by employing local instead of global observables (see Ref. [26]).

The parameter optimization loop results in an approximately consistent family, \mathcal{F} , of histories, where the consistency parameter ϵ is upper bounded in terms of the final cost (see Methods). In Step **c** in Fig. 2, we then generate the probabilities for the most likely histories by repeatedly preparing σ^A and measuring in the standard basis, where the measurement frequencies give the probabilities. Step **e** shows how one prepares the set of projection operators for any given history in \mathcal{F} . These projectors can then be characterized with an efficient number of observables (i.e., avoiding full state tomography) to learn important information about the histories.

Let us discuss the scaling of VCH. With the potential exceptions of the Hamiltonian evolution and the projection operators, the complexity of our quantum circuits (i.e., the gate count, circuit depth, and total number of required qubits) scales linearly with both the system size n and the number of times k considered. The complexity of Hamiltonian evolution to some accuracy is problem dependent, but we typically expect polynomial scaling in n for physical systems with properties like translational

symmetry [34]. On the other hand, we consider the circuit depth for preparing the history projectors to be a refinement parameter. One can begin with a short-depth ansatz for the projectors and incrementally increase the depth to refine the ansatz, potentially improving the approximate consistency. We therefore expect the overall scaling of our quantum circuits to be polynomial in n and k for the anticipated use cases of VCH.

The complexity of minimizing our non-convex cost function is unknown, which is typical for VHQCA. As classical methods for finding consistent families also involve optimizing over some parameterization for the projectors, classical methods also need to deal with this optimization complexity issue.

While the number of required repetitions of the probability readout step can scale inefficiently in n and k for certain families of histories, we assume that minimizing the cost outputs a family \mathcal{F} for which the probability readout step is efficient. (See Methods for elaboration on this point.)

This scaling behavior means that for systems that can be tractably simulated on a quantum computer and whose properties of interest are simple to implement, we achieve an exponential speedup and reduction in the needed resources as compared to classical approaches to this problem.

EXPERIMENTAL IMPLEMENTATIONS

Quantum Hardware Spin in a Magnetic Field

We now present an experimental demonstration of VCH on a cloud quantum computer. See the SM for further details on this implementation. We examine the two time histories of a spin-1/2 particle in a magnetic field \$B\hat{z}\$, whose Hamiltonian is \$H = -\gamma B\sigma_z\$. The histories we consider have a time step \$\Delta t\$ between the initial state (chosen to be \$\rho = |+\rangle\langle +|\$, with \$|+\rangle = 1/\sqrt{2}(|0\rangle + |1\rangle)\$ ) and first projector, as well as between the first and second projector, chosen so that \$\gamma B\Delta t = 2\text{rad}\$. Additionally, we only consider projectors onto the \$xy\$ plane of the Bloch sphere, parameterized by their azimuth. For this model, Fig. 3 shows the landscape of the cost in (9) for ~~a simulator and for~~ the ibmqx5 quantum processor [35] as well as a simulator whose precision was limited by imposing the same finite statistics as were collected with the quantum processor. Several minima found by running VCH on ibmqx5 are superimposed on the landscape. ~~These (All points found below a noise threshold were considered to be equally valid minima.)~~ As these minima correspond reasonably well to theoretically consistent families, ~~and hence~~ this represents a successful proof-of-principle implementation of VCH.

FIG. 3. Consistency of three-time histories for a spin-1/2 particle in a magnetic field, with initial state $\rho = |+\rangle\langle +|$. The full-trace cost landscape, $C(\phi_1, \phi_2)$, is plotted as a function of the azimuths, ϕ_1 and ϕ_2 , of the first and second projection bases, which we constrained to the xy plane of the Bloch sphere. The point $(0, 0)$ corresponds to both projections being along the x axis. Consistency is expected everywhere along certain vertical lines ($\phi_1 = 2 + n\pi$ rad), as they correspond to histories where the initial state is one of the projectors after the first time step, so there are no branches to interfere in the second time step. In addition, some slope-one lines ($\phi_2 = \phi_1 + (2 + n\pi)$ rad) should be consistent, as they correspond to histories where the second projectors are the same as the first after time evolution, so no interference occurs in the second time step. Indeed, one can see valleys in the cost landscapes for these vertical and slope-one lines, when the cost is quantified on a simulator **a** and on the ibmqx5 quantum computer **b**. Note that negative cost values are possible due to finite statistics. The white “x” symbols in **b** mark some of the non-unique minima that the VCH algorithm found.

Simulator Chiral Molecule

To highlight applications that will be possible on future hardware, we simulate VCH to observe the quantum-to-classical transition for a chiral molecule [36, 37]. It has

FIG. 4. The cost landscape for stationary histories of the chiral molecule. Since the projectors in these stationary histories are always along a single axis, we plot the cost on points where this axis would intersect the surface of the Bloch sphere. The bottom row of spheres are the same as the top, but rotated for additional perspective. Panels **a** and **b** show the full and partial trace cost functions, respectively, for the case where the environment interactions are negligible ($\theta_z = 5$ rad, $\theta_x = .01$ rad), and thus we find that the energy eigenbasis (z axis) is the only consistent stationary family as all others will branch as they evolve. In contrast, panels **c** and **d** are the full and partial trace cost functions, respectively, for the case where the environment interactions dominate ($\theta_z = .01$ rad, $\theta_x = 5$ rad). One can see in **c** and **d** a significant difference between the full and partial trace costs for the y axis, meaning that this family of histories is consistent but not classical. In this regime, we also see that the chirality basis (the x axis) is a local minimum for both cost functions and thus is approximately consistent and classical. For this chirality basis family, there is a $\sim 0.01\%$ chance that the molecule will change chirality during the evolution, showing that the quantum-to-classical transition leaves this system in a stabilized chiral state.

been modeled as a two level system where the right $|R\rangle$ and left $|L\rangle$ chirality states are described as $|R\rangle/|L\rangle = |+\rangle/|-\rangle = \frac{1}{\sqrt{2}}(|0\rangle \pm |1\rangle)$ [37]. A chiral molecule in isolation would tunnel between $|R\rangle$ and $|L\rangle$, but we consider the molecule to be in a gas, where collisions with other molecules convey information about the molecule's chirality to its environment. This information transfer is modeled by a rotation by angle θ_x about the x axis of an environment qubit, controlled on the system's chirality, and for simplicity we suppose such collisions are evenly spaced at five points in time. (See the SM for further details.) We then consider simple families of stationary histories [37], where the projector set corresponds to the same basis at all five times (just after a collision occurs). Letting θ_z be the precession angle due to tunneling in the time between collisions, we can then explore the competition between decoherence and tunneling. Figure 4 shows our results for this model. Notably we observe the transition from a quantum regime, where the chirality is not consistent, to a classical regime, where the chirality is both consistent and stable over time.

DISCUSSION

Making it possible to apply the tools and concepts of quantum foundations to a wide array of physical

situations, as VCH will, is an important step for our understanding of the physical world. Specifically by providing an exponential speedup and reduction in resources over classical methods, VCH will provide a way to study phenomena including the quantum-to-classical transition [31, 32, 38], dynamics of quantum phase transitions [39], quantum biological processes [40], conformational changes [41], and many other complex phenomena that so far have been computationally intractable. In addition, VCH could be applied to study quantum algorithms themselves [42]. In order to highlight such potential applications and examine their resource requirements, we now focus on two of them: the emergence of classical diffusive dynamics in quantum spin systems and the appearance of defined pathways in protein folding.

In the context of Nuclear Magnetic Resonance (NMR) experiments, it has long been known that systems with many spins obey a classical diffusion equation while smaller spin systems undergo Rabi oscillations. Despite the long history of spin diffusion studies [43-45], there is still no derivation of the transition from quantum oscillations to classical diffusion that can predict the size of the system where we should find that transition, or the nature of the transition. Applying VCH to the study of histories of spin systems would clarify this point by showing the scale and abruptness with which the diffusive behavior emerges. Since spin diffusion has been observed

for systems as small as $\sim 30,000$ spins [46], we estimate that between $\sim 10^2$ and $\sim 10^3$ qubits would allow us to study this transition. For more details about this application, see the SM.

In the protein folding community there are currently two main schools of thought on how proteins fold. The first is that proteins fold along well determined pathways with discrete folding units (foldons) [47], while the second is that there should be a funnel in the energy landscape of folding configurations, causing the system to explore a wide range of configurations before settling into the final state [48]. The deterministic pathways of the foldon model are favored by NMR experiments, raising the question of whether these views can be reconciled [47]. By providing the means to study the dynamic emergence of classical paths, i.e., the quantum-to-classical transition for proteins, VCH could resolve this discrepancy. For this purpose, we estimate that between $\sim 10^3$ to $\sim 10^4$ qubits will be needed. See the SM for more details on this application and resource estimate.

Finally, our work highlights the synergy of two distinct fields, quantum foundations and quantum computational algorithms, and hopefully will inspire further research into their intersection.

METHODS

A. Evaluation of the Cost

Figure 5 shows the circuits for computing the full trace cost (partial trace cost) from two copies of σ^A (σ^{SA}). Note that both costs can be written as a difference of purities:

$$C = \text{Tr}((\sigma^A)^2) - \text{Tr}(\mathcal{Z}^A(\sigma^A)^2) \quad (11)$$

$$C_{\text{pt}} = \text{Tr}((\sigma^{SA})^2) - \text{Tr}(\mathcal{Z}^A(\sigma^{SA})^2). \quad (12)$$

The $\text{Tr}((\sigma^A)^2)$ and $\text{Tr}((\sigma^{SA})^2)$ terms are computed via the Swap Test, with a depth-two circuit and classical post-processing that scales linearly in the number of qubits [49, 50]. A similar but even simpler circuit, called the Diagonalized Inner Product (DIP) Test [26], calculates the $\text{Tr}(\mathcal{Z}^A(\sigma^A)^2)$ term with a depth one circuit and no post-processing. Finally, the $\text{Tr}(\mathcal{Z}^A(\sigma^{SA})^2)$ term is evaluated with the Partial-DIP (PDIP) Test [26], a depth-two circuit that is a hybridization of the Swap Test and the DIP Test.

B. Precision of probability readout

One does not know *a priori* how many histories will be characterized in the probability readout step (Fig. 2c). Due to statistical noise, the probability of histories with greater probability will be determined with greater relative precision than those with lesser probability. Hence, it is reasonable to set a precision (or statistical noise)

FIG. 5. Circuits for computing the full trace cost (panel a) and the partial trace cost (panel b).

threshold, ϵ_{max} . Let N_{readout} be the number of repetitions of the probability readout circuit. Then, histories \mathcal{Y}^α whose bitstring α occurs with frequency $f_\alpha < \sqrt{N_{\text{readout}}}/\epsilon_{\text{max}}$ should be ignored, since their probabilities $p(\alpha) = f_\alpha/N_{\text{readout}}$ were not characterized with the desired precision. We separate \mathcal{F} into the set \mathcal{F}_c of histories whose probabilities are above the precision threshold (which we previously referred to loosely as the most likely histories), and the set of all other histories in \mathcal{F} :

$$\mathcal{F} = \mathcal{F}_c \cup \overline{\mathcal{F}_c}. \quad (13)$$

Computational complexity can be hidden in the value of N_{readout} needed to obtain a desired precision for the probabilities of histories of interest. This issue is closely connected to the entropy of the set $\{\mathcal{D}(\alpha, \alpha)\}$, or equivalently, the entropy of the quantum state $\mathcal{Z}^A(\sigma^A)$. When $\mathcal{Z}^A(\sigma^A)$ is high entropy, an exponentially large number of histories may have non-zero probability, and hence N_{readout} would need to grow exponentially. VCH is therefore better suited to applications where there is a small subset of the histories that are far more probable than the rest. In the parameter optimization loop, one can select for families with this property by penalizing families for which $\mathcal{Z}^A(\sigma^A)$ has high entropy. Specifically, by noting that $P := \text{Tr}(\mathcal{Z}^A(\sigma^A)^2)$ can be efficiently computed via the circuit in Fig. 5a, one can modify the costs functions in Eq. (9) and Eq. (10) to be $\tilde{C} = C/P$ and $\tilde{C}_{\text{pt}} = C_{\text{pt}}/P$.

We remark that classicality is intimately connected to predictability, with the emergence of classicality linked to the so-called predictability sieve [51, 52]. Since the CH formalism is typically used to find classical families, this implies predictable families (i.e., families with low entropy or high purity P) are arguably of the most interest. Hence, our modified cost function \tilde{C} also serves to select those consistent families with histories that are the most predictable, and therefore the most classical.

C. Approximate Consistency

Here we discuss how VCH outputs an upper bound on the consistency parameter ϵ . Let us first relate the cost C to ϵ . For any pair of histories \mathcal{Y}^α and $\mathcal{Y}^{\alpha'}$ in \mathcal{F} ,

$$|\mathcal{D}(\alpha, \alpha')|^2 \leq C/2, \quad (14)$$

which follows from Eq. (9) and the fact that $|\mathcal{D}(\alpha, \alpha')| = |\mathcal{D}(\alpha', \alpha)|$. Let us define

$$\epsilon_{\alpha, \alpha'} := \sqrt{\frac{C}{2\mathcal{D}(\alpha, \alpha)\mathcal{D}(\alpha', \alpha')}}. \quad (15)$$

Then it follows from Eq. (14) that

$$|\mathcal{D}(\alpha, \alpha')|^2 \leq \epsilon_{\alpha, \alpha'}^2 \mathcal{D}(\alpha, \alpha)\mathcal{D}(\alpha', \alpha'), \quad (16)$$

which corresponds to the approximate consistency condition from Eq. (6). Hence, probability sum rules for these two histories are satisfied within error $\epsilon_{\alpha, \alpha'}$, which can be calculated from Eq. (15) for histories in \mathcal{F}_c since the probabilities are known for these histories.

Next, consider histories in $\overline{\mathcal{F}}_c$. As we do not have enough information to differentiate these histories, we advocate combining the elements of $\overline{\mathcal{F}}_c$ into a single coarse-grained history \mathcal{Y}^γ .

Let \mathcal{Y}^β be the least likely history in \mathcal{F}_c . Then defining $\delta^2 = \mathcal{D}(\gamma, \gamma)/\mathcal{D}(\beta, \beta)$, we can make use of the positive semi-definite property of σ^A to write:

$$|\mathcal{D}(\gamma, \beta)|^2 \leq \mathcal{D}(\gamma, \gamma)\mathcal{D}(\beta, \beta) = \delta^2\mathcal{D}(\beta, \beta)^2. \quad (17)$$

Since \mathcal{Y}^β is the least likely history in \mathcal{F}_c , this expression then lets us bound the error on the probability sum rule (giving a weaker approximate consistency condition [30]) between \mathcal{Y}^γ and any $\mathcal{Y}^\alpha \in \mathcal{F}_c$ as:

$$\begin{aligned} |\mathcal{D}(\gamma, \alpha)| &\leq \delta\mathcal{D}(\alpha, \alpha) \\ &\leq \delta(\mathcal{D}(\gamma, \gamma) + \mathcal{D}(\alpha, \alpha)). \end{aligned} \quad (18)$$

It is then possible to characterize the approximate consistency of the histories of \mathcal{F} pairwise with $\epsilon_{\alpha, \alpha'}$ and δ . Alternatively, to give an upper bound on the overall consistency ϵ , we take the greatest of these pairwise bounds:

$$\epsilon \leq \max(\{\epsilon_{\alpha, \alpha'}\} \cup \{\delta\}). \quad (19)$$

For those applications where we are working with the partial trace consistency, the notion of approximate consistency is somewhat more obscured. In order to generate probabilities and bound ϵ , we therefore recommend evaluating the full trace cost function at the minimum found with the partial trace cost. This approach is helpful since any partial trace consistent family will also be full trace consistent and the partial trace consistency does not directly allow one to discuss probabilities in the same way. Taking this approach allows us to then directly utilize the approximate consistency framework above.

ACKNOWLEDGMENTS

We thank IBM for the use of their quantum processor. The views expressed in this article are those of the authors and not of IBM. This work was supported by the U.S. Department of Energy (DOE), Office of Science, Office of High Energy Physics, and also by the U.S. DOE, Office of Science, Basic Energy Sciences, Materials Sciences and Engineering Division, Condensed Matter Theory Program. All authors acknowledge support from the LDRD program at Los Alamos National Laboratory (LANL). LC was also supported by the DOE through the J. Robert Oppenheimer fellowship. ATS and PJC additionally acknowledge support from the LANL ASC Beyond Moore's Law project. Finally, WHZ acknowledges partial support by the Foundational Questions Institute grant FQXi-1821, and Franklin Fetzner Fund, a donor advised fund of the Silicon Valley Community Foundation.

AUTHOR CONTRIBUTIONS

All authors contributed to the preparation and revision of the manuscript. P.J.C. invented the algorithm and developed the basic formalism. A.A. designed and carried out the experimental implementations, analyzed the results, and contributed to the formalism. L.C., A.T.S., and W.H.Z. consulted on all stages of the project.

COMPETING INTERESTS

The authors declare no competing interests.

DATA AVAILABILITY

The data used to create the figures in this article are available upon request. Requests should be sent to the corresponding author.

[1] J. A. Wheeler and W. H. Zurek, editors. *Quantum Theory and Measurement (Princeton Series in Physics)*. Princeton University Press, 2016.

[2] G. Auletta. *Foundations and Interpretation of Quantum Mechanics*. World Scientific, 2000.

- [3] E. Joos and H. D. Zeh. The emergence of classical properties through interaction with the environment. *Zeitschrift für Physik B Condensed Matter*, 59(2):223–243, 1985.
- [4] W. H. Zurek. Decoherence, einselection, and the quantum origins of the classical. *Rev. Mod. Phys.*, 75:715–775, 2003.
- [5] M. A. Schlosshauer. *Decoherence: and the quantum-to-classical transition*. Springer Science & Business Media, 2007.
- [6] R. B. Griffiths. Consistent histories and the interpretation of quantum mechanics. *Journal of Statistical Physics*, 36(1):219–272, 1984.
- [7] R. Omnès. Logical reformulation of quantum mechanics. I. foundations. *Journal of Statistical Physics*, 53(3):893–932, 1988.
- [8] M. Gell-Mann and J. B. Hartle. Quantum Mechanics in the Light of Quantum Cosmology. In *Proceedings of the 3rd International Symposium Foundations of Quantum Mechanics in the Light of New Technology*, 1989.
- [9] J. Hartle and T. Hertog. One bubble to rule them all. *Phys. Rev. D*, 95:123502, 2017.
- [10] S. Lloyd. Decoherent histories approach to the cosmological measure problem. *arXiv:1608.05672*.
- [11] T. A. Brun. Quantum jumps as decoherent histories. *Phys. Rev. Lett.*, 78:1833–1837, 1997.
- [12] J. J. Halliwell and J. M. Yearsley. Quantum arrival time formula from decoherent histories. *Physics Letters A*, 374(2):154 – 157, 2009.
- [13] J. J. Halliwell and J. M. Yearsley. Arrival times, complex potentials, and decoherent histories. *Phys. Rev. A*, 79:062101, 2009.
- [14] C. Anastopoulos and N. Savvidou. Time of arrival and localization of relativistic particles. *arXiv:1807.06533*.
- [15] Todd A. Brun. Quasiclassical equations of motion for nonlinear brownian systems. *Phys. Rev. D*, 47:3383–3393, Apr 1993.
- [16] H.-J. Pohle. How to calculate decoherence matrices numerically. *Physica A: Statistical Mechanics and its Applications*, 213(3):435 – 449, 1995.
- [17] D. Schmidtke and J. Gemmer. Numerical evidence for approximate consistency and markovianity of some quantum histories in a class of finite closed spin systems. *Phys. Rev. E*, 93:012125, 2016.
- [18] J. Preskill. Quantum Computing in the NISQ era and beyond. *Quantum*, 2:79, 2018.
- [19] A. Peruzzo, J. McClean, P. Shadbolt, M.-H. Yung, X.-Q. Zhou, P. J. Love, A. Aspuru-Guzik, and J. L. O’Brien. A variational eigenvalue solver on a photonic quantum processor. *Nature Communications*, 5:4213, 2014.
- [20] E. R. Anschuetz, J. P. Olson, A. Aspuru-Guzik, and Y. Cao. Variational quantum factoring. *arXiv:1808.08927*.
- [21] E. Farhi, J. Goldstone, and S. Gutmann. A quantum approximate optimization algorithm. *arXiv:1411.4028*.
- [22] J. Romero, J. P. Olson, and A. Aspuru-Guzik. Quantum autoencoders for efficient compression of quantum data. *Quantum Science and Technology*, 2(4):045001, 2017.
- [23] Y. Li and S. C. Benjamin. Efficient variational quantum simulator incorporating active error minimization. *Physical Review X*, 7(2):021050, 2017.
- [24] P. D. Johnson, J. Romero, J. Olson, Y. Cao, and A. Aspuru-Guzik. QVECTOR: an algorithm for device-tailored quantum error correction. *arXiv:1711.02249*.
- [25] S. Khatri, R. LaRose, A. Poremba, L. Cincio, A. T. Sornborger, and P. J. Coles. Quantum assisted quantum compiling. *arXiv:1807.00800*.
- [26] R. LaRose, A. Tikku, È. O’Neel-Judy, L. Cincio, and P. J. Coles. Variational quantum state diagonalization. *arXiv:1810.10506*.
- [27] R. B. Griffiths. *Consistent Quantum Theory*. Cambridge University Press, 2001.
- [28] J. J. Halliwell. A review of the decoherent histories approach to quantum mechanics. *Annals of the New York Academy of Sciences*, 755(1):726–740, 1995.
- [29] P. C. Hohenberg. Colloquium: An introduction to consistent quantum theory. *Rev. Mod. Phys.*, 82:2835–2844, 2010.
- [30] H. F. Dowker and J. J. Halliwell. Quantum mechanics of history: The decoherence functional in quantum mechanics. *Phys. Rev. D*, 46:1580–1609, 1992.
- [31] C. J. Riedel, W. H. Zurek, and M. Zwolak. Objective past of a quantum universe: Redundant records of consistent histories. *Phys. Rev. A*, 93:032126, 2016.
- [32] J. Finkelstein. Definition of decoherence. *Phys. Rev. D*, 47:5430–5433, 1993.
- [33] J. R. McClean, J. Romero, R. Babbush, and A. Aspuru-Guzik. The theory of variational hybrid quantum-classical algorithms. *New Journal of Physics*, 18(2):023023, 2016.
- [34] D. W. Berry, A. M. Childs, R. Cleve, R. Kothari, and R. D. Somma. Simulating hamiltonian dynamics with a truncated taylor series. *Phys. Rev. Lett.*, 114:090502, 2015.
- [35] IBM Q 16 Rueschlikon backend specification. <https://github.com/Qiskit/qiskit-backend-information/tree/master/backends/rueschlikon/V1>, 2018.
- [36] J. Trost and K. Hornberger. Hund’s paradox and the collisional stabilization of chiral molecules. *Phys. Rev. Lett.*, 103:023202, 2009.
- [37] P. J. Coles, V. Gheorghiu, and R. B. Griffiths. Consistent histories for tunneling molecules subject to collisional decoherence. *Phys. Rev. A*, 86:042111, 2012.
- [38] J. P. Paz and W. H. Zurek. Environment-induced decoherence, classicality, and consistency of quantum histories. *Phys. Rev. D*, 48:2728–2738, 1993.
- [39] W. H. Zurek, U. Dorner, and P. Zoller. Dynamics of a Quantum Phase Transition. *Phys. Rev. Lett.*, 95:105701, 2005.
- [40] M. Allegra, P. Giorda, and S. Lloyd. Global coherence of quantum evolutions based on decoherent histories: Theory and application to photosynthetic quantum energy transport. *Phys. Rev. A*, 93:042312, 2016.
- [41] H. Liu, M. Elstner, E. Kaxiras, T. Frauenheim, J. Hermans, and W. Yang. Quantum mechanics simulation of protein dynamics on long timescale. *Proteins*, 44(4):484–489, 2001.
- [42] D. Poulin. Classicality of quantum information processing. *Phys. Rev. A*, 65:042319, 2002.
- [43] N. Bloembergen. On the interaction of nuclear spins in a crystalline lattice. *Physica*, 15(3):386 – 426, 1949.
- [44] I. J. Lowe and S. Gade. Density-matrix derivation of the spin-diffusion equation. *Phys. Rev.*, 156:817–825, Apr 1967.
- [45] Jean-Nicolas Dumez. *Many body dynamics in nuclear spin diffusion*. Theses, Ecole normale supérieure de lyon - ENS LYON, July 2011.

- [46] S. Adachi, R. Kaji, S. Furukawa, Y. Yokoyama, and S. Muto. Nuclear spin depolarization via slow spin diffusion in single InAlAs quantum dots observed by using erase-pump-probe technique. *Journal of Applied Physics*, 111(10):103531, 2012.
- [47] S. Walter Englander and Leland Mayne. The case for defined protein folding pathways. *Proceedings of the National Academy of Sciences*, 114(31):8253–8258, 2017.
- [48] William A. Eaton and Peter G. Wolynes. Theory, simulations, and experiments show that proteins fold by multiple pathways. *Proceedings of the National Academy of Sciences*, 114(46):E9759–E9760, 2017.
- [49] J. C. Garcia-Escartin and P. Chamorro-Posada. Swap test and Hong-Ou-Mandel effect are equivalent. *Physical Review A*, 87(5):052330, 2013.
- [50] L. Cincio, Y. Subaşı, A. T. Sornborger, and P. J. Coles. Learning the quantum algorithm for state overlap. *New Journal of Physics*, 20(11):113022, 2018.
- [51] Wojciech H. Zurek. Preferred States, Predictability, Classicality and the Environment-Induced Decoherence. *Progress of Theoretical Physics*, 89(2):281–312, 02 1993.
- [52] D. A. R. Dalvit, J. Dziarmaga, and W. H. Zurek. Predictability sieve, pointer states, and the classicality of quantum trajectories. *Phys. Rev. A*, 72:062101, Dec 2005.

Supplementary Material for “Variational Consistent Histories: A Hybrid Algorithm for Quantum Foundations”

Appendix A: Generalizations

Here we discuss various generalizations of the circuits shown in the main text, which presented our VCH algorithm for the special case of branch-independent histories of a one-qubit system S with no environment E .

1. Multi-Qubit Systems

The circuits in the main text showed systems S composed of a single qubit. The generalization to multi-qubit systems is straightforward. We must discuss the generalizations of both the state preparation circuit in Fig. 2 as well as the cost evaluation circuits in Fig. 5.

Figure S.1 illustrates how the state preparation circuit generalizes to multi-qubit systems. In particular, this figure shows how a portion of state preparation circuit (the portion that entangles the system to the ancillas) generalizes for the case of a fine-grained set of projectors. (Note that the case of a coarse-grained set of projectors is discussed in the next subsection.)

The cost evaluation circuits in Fig. 5 generalize as follows. For fine-grained histories, one needs n ancillas for each time step and hence a total of nk ancillas. The circuits in Fig. 5 shown for k ancillas generalize in a straightforward way, where now one has nk ancilla systems. In addition, the circuits in Fig. 5b also involve the S system, and hence all n qubits in S must be included in this circuit. Again, these n qubits are included in the most straightforward way (in the same way that the single qubit S system appears in the circuits in Fig. 5b).

FIG. S.1. The generalization of our state preparation circuit to multi-qubit systems S . In this example, we show the portion of the circuit that entangles the system and the ancillas, for the special case of a fine-grained set of projectors. In this fine-grained case, one employs the same number of ancilla qubits as are in S , i.e., n qubits.

2. Coarse Grained Histories

Multi-qubit systems S allow for non-trivial coarse-grained histories. In such families of histories, the sets P_j are composed of projectors whose ranks are possibly greater than one. We remark that coarse-grained histories are often important to the study of macroscopic systems and the quantum-to-classical transition. VCH can easily be adapted to study coarse-grained histories as follows.

For each time t_j , one should decide (prior to running VCH) ~~what~~ projector ranks that one is interested in. VCH will then optimize over sets of projectors with these particular ranks. The projector ranks can therefore be viewed as hyperparameters, i.e., parameters that one fixes for a given run of VCH.

For instance, suppose S is composed of a pair of spins. In this case, Fig. S.2 shows two examples of the state preparation circuit for a single time step. In the first example, Fig. S.2a, we consider a projector set that contains two rank-two projectors revealing whether the spins were aligned or anti-aligned. In the second example, Fig. S.2b, we consider a projector set that contains a rank-three and a rank-one projector that respectively indicate whether the spins are in the triplet states or the the singlet state. Note that the ranks of the projectors are determined by the gate that entangles the system to the ancilla, which is a single CNOT gate in Fig. S.2a and a Toffoli gate in Fig. S.2b. Hence the choice of the projector ranks (mentioned in the previous paragraph) translates into a choice of gate sequence that entangles the system to the ancilla.

FIG. S.2. Examples of implementing coarse-grained projector sets in our state preparation circuit, when S corresponds to two spin-1/2 particles. The projectors in **a** record whether the two spins are aligned or anti-aligned, while the projectors in **b** differentiate between the spin singlet and spin triplet states.

3. Nontrivial Environments

For many applications of VCH, (e.g., the chiral molecule example in the main text) it will be helpful to explicitly model an environment E . We can think of this case as a particular choice of coarse graining where the projectors we consider only act on a subsystem of our

model (the S system) and do not directly record any information about E . Note that the Hamiltonian evolution involves both S and E , as shown in Fig. S.3.

FIG. S.3. Simple example with an environment E . The projectors still only act on S , but the evolution includes both S and E .

4. Branch Dependent Histories

A final generalization that we consider are families of branch dependent histories [1], or histories where the projector set at a given time may depend on the properties of the system at earlier points in the histories. VCH can accommodate these histories, as follows.

The basic idea is that the unitary gate B_j that determines the projector set at time t_j now becomes a controlled unitary. Specifically, the control system(s) for B_j are (potentially) all the ancilla qubits associated with times $t_i < t_j$. So the choice of projector set at some time is influenced by the ancilla states for earlier times.

Figure S.4 shows an example of what this looks like, for the special case of only two times. In this figure, if the first ancilla is in the $|0\rangle$ state ($|1\rangle$ state), then the B_2 unitary ($B'_2 B_2$ unitary) is applied at the second time step. For more general cases, the B'_2 unitary shown here **might would** be replaced by a sequence of controlled unitaries controlled by different ancilla qubits.

FIG. S.4. Example implementation of a branch dependent projector set in our state preparation circuit. In this circuit, depending upon the result for t_1 , either B_2 or the product $B'_2 B_2$ defines the projector set for the second time.

Appendix B: Generalized state preparation

We now present the details of our generalized state preparation circuit (as shown in Fig. S.5) and show that σ^{SA} and σ^{A} have the properties we claim in the main text. Note that our treatment here includes all of the generalizations discussed above in Appendix A. We begin with the input state $\rho^{\text{SE}} \otimes |\mathbf{0}\rangle\langle\mathbf{0}|^{\text{A}}$ (where the superscript SE denotes the system and its environment and A

denotes the ancillas). We then apply the gate sequence associated with the P_1 projector set, which includes B_1 , a multi-qubit gate that entangles S and A (which we refer to as the “entangling gate”), and then B_1^\dagger . This gives the state:

$$\sum_{\alpha_1, \alpha'_1} \left[P_1^{\alpha_1} \rho^{\text{SE}} P_1^{\alpha'_1 \dagger} \right] \otimes \left[|\alpha_1\rangle\langle\alpha'_1| \otimes |\mathbf{0}\rangle\langle\mathbf{0}| \right]^{\text{A}}. \quad (\text{B1})$$

Note that the system and ancilla are (possibly) entangled at this point.

Next in our state preparation circuit is the time evolution from t_1 to t_2 , given by $e^{-iH\Delta t_{1,2}}$. This is followed by the gate sequence associated with P_2 , which in general may be branch dependent. The resulting state is

$$\sum_{\alpha_1, \alpha'_1, \alpha_2, \alpha'_2} \left[P_2^{\alpha_2}(\alpha_1) e^{-iH\Delta t_{1,2}} P_1^{\alpha_1} \rho^{\text{SE}} P_1^{\alpha'_1 \dagger} e^{iH\Delta t_{1,2}} P_2^{\alpha'_2 \dagger}(\alpha_1) \right] \otimes \left[|\alpha_1\rangle\langle\alpha'_1| \otimes |\alpha_2\rangle\langle\alpha'_2| \otimes |\mathbf{0}\rangle\langle\mathbf{0}| \right]^{\text{A}}, \quad (\text{B2})$$

where the notation $P_2^{\alpha_2}(\alpha_1)$ indicates that the second projector set depends on α_1 . Repeating this state evolution until we have applied the gate sequences associated with all k projector sets (and switching to the Heisenberg picture), we end up with

$$\begin{aligned} & \sum_{\alpha, \alpha'} \left[P_k^{\alpha_k}(t_k) \dots P_2^{\alpha_2}(t_2) P_1^{\alpha_1}(t_1) \rho^{\text{SE}} \right. \\ & \quad \left. P_1^{\alpha'_1}(t_1)^\dagger P_2^{\alpha'_2}(t_2)^\dagger \dots P_k^{\alpha'_k}(t_k)^\dagger \right] \\ & \quad \otimes \left[(|\alpha_1\rangle\langle\alpha'_1|) \otimes (|\alpha_2\rangle\langle\alpha'_2|) \otimes \dots \otimes (|\alpha_k\rangle\langle\alpha'_k|) \right]^{\text{A}} \\ & = \sum_{\alpha, \alpha'} \mathcal{C}^\alpha \rho^{\text{SE}} \mathcal{C}^{\alpha' \dagger} \otimes (|\alpha\rangle\langle\alpha'|)^{\text{A}} \end{aligned} \quad (\text{B3})$$

Note that we have suppressed explicit branch dependence here to simplify notation. Branch dependence does not alter the formalism except to make the later projectors functions of the earlier α_i 's, so our treatment remains fully general.

If we then trace out the environment (which in the circuit means not measuring it) we are then left with σ^{SA} :

$$\sigma^{\text{SA}} = \sum_{\alpha, \alpha'} \text{Tr}_E(\mathcal{C}^\alpha \rho^{\text{SE}} \mathcal{C}^{\alpha' \dagger}) \otimes (|\alpha\rangle\langle\alpha'|)^{\text{A}}. \quad (\text{B4})$$

By examining Eq. (B4), we can see that $(\mathbb{1} \otimes |\alpha\rangle)\sigma^{\text{SA}}(\mathbb{1} \otimes \langle\alpha|)$ is precisely $\mathcal{D}_{\text{pt}}(\alpha, \alpha') = \text{Tr}_E(\mathcal{C}^\alpha \rho^{\text{SE}} \mathcal{C}^{\alpha' \dagger})$. Further, if we similarly trace over the system S , we get:

$$\sigma^{\text{A}} = \sum_{\alpha, \alpha'} \text{Tr}(\mathcal{C}^\alpha \rho^{\text{SE}} \mathcal{C}^{\alpha' \dagger}) (|\alpha\rangle\langle\alpha'|)^{\text{A}}. \quad (\text{B5})$$

We can thus see that we have prepared a density matrix whose elements are $\mathcal{D}(\alpha, \alpha') = \text{Tr}(\mathcal{C}^\alpha \rho^{\text{SE}} \mathcal{C}^{\alpha' \dagger})$, as claimed in the main text.

FIG. S.5. The generalized state preparation circuit. A similar circuit is included as part of the flowchart, but this version incorporates larger systems and branch dependence explicitly. The multiqubit gate $\begin{array}{c} \square \\ \oplus \end{array}$ denotes a set of entangling gates controlled on the standard basis of the system qubits, while $\begin{array}{c} \square \\ \square \end{array}$ represents a parameterized unitary acting on the system controlled on the standard basis of the ancilla qubits.

Appendix C: Derivation of Cost Functions

1. Full trace cost

Let us now derive the equivalence stated in the definition of our full trace cost function, Eq. (9). Starting with the definition of C we have:

$$\begin{aligned}
C &:= \sum_{\alpha \neq \alpha'} |\mathcal{D}(\alpha, \alpha')|^2 \\
&= \sum_{\alpha \neq \alpha'} \langle \alpha | \sigma^A | \alpha' \rangle \langle \alpha' | \sigma^A | \alpha \rangle \\
&= \sum_{\alpha \neq \alpha'} \text{Tr} ((|\alpha\rangle\langle\alpha|) \sigma^A (|\alpha'\rangle\langle\alpha'|) \sigma^A) \\
&= \sum_{\alpha, \alpha'} \text{Tr} ((|\alpha\rangle\langle\alpha|) \sigma^A (|\alpha'\rangle\langle\alpha'|) \sigma^A) \\
&\quad - \sum_{\alpha} \text{Tr} ((|\alpha\rangle\langle\alpha|) \sigma^A (|\alpha\rangle\langle\alpha|) \sigma^A) \\
&= \text{Tr}((\sigma^A)^2) - \text{Tr}(\mathcal{Z}^A(\sigma^A)^2) \\
&= D_{\text{HS}}(\sigma^A, \mathcal{Z}^A(\sigma^A)). \tag{C1}
\end{aligned}$$

Therefore, the circuits we use to calculate $\text{Tr}((\sigma^A)^2)$ and $\text{Tr}(\mathcal{Z}^A(\sigma^A)^2)$ implement this cost function as claimed.

2. Partial trace cost

Arriving at the expression for the partial trace cost function (Eq. (10)) is similar if slightly more complicated:

$$\begin{aligned}
C_{\text{pt}} &:= \sum_{\alpha \neq \alpha'} \|\mathcal{D}_{\text{pt}}(\alpha, \alpha')\|_{\text{HS}}^2 \\
&= \sum_{\alpha \neq \alpha'} \text{Tr}_{\text{S}} (\mathcal{D}_{\text{pt}}(\alpha, \alpha') \mathcal{D}_{\text{pt}}(\alpha, \alpha')^\dagger) \\
&= \sum_{\alpha \neq \alpha'} \text{Tr}_{\text{S}} ((\mathbb{1} \otimes \langle \alpha |) (\mathbb{1} \otimes | \alpha \rangle \langle \alpha |) \sigma^{\text{SA}} \\
&\quad (\mathbb{1} \otimes | \alpha' \rangle \langle \alpha' |) \sigma^{\text{SA}} (\mathbb{1} \otimes | \alpha \rangle)) \\
&= \sum_{\alpha \neq \alpha'} \text{Tr} ((\mathbb{1} \otimes | \alpha \rangle \langle \alpha |) \sigma^{\text{SA}} (\mathbb{1} \otimes | \alpha' \rangle \langle \alpha' |) \sigma^{\text{SA}}) \\
&= \sum_{\alpha, \alpha'} \text{Tr} ((\mathbb{1} \otimes | \alpha \rangle \langle \alpha |) \sigma^{\text{SA}} (\mathbb{1} \otimes | \alpha' \rangle \langle \alpha' |) \sigma^{\text{SA}}) \\
&\quad - \sum_{\alpha} \text{Tr} ((\mathbb{1} \otimes | \alpha \rangle \langle \alpha |) \sigma^{\text{SA}} (\mathbb{1} \otimes | \alpha \rangle \langle \alpha |) \sigma^{\text{SA}}) \\
&= \text{Tr}((\sigma^{\text{SA}})^2) - \text{Tr}(\mathcal{Z}^A(\sigma^{\text{SA}})^2) \\
&= D_{\text{HS}}(\sigma^{\text{SA}}, \mathcal{Z}^A(\sigma^{\text{SA}})). \tag{C2}
\end{aligned}$$

As with the full trace cost function, the circuits we use to calculate $\text{Tr}((\sigma^{\text{SA}})^2)$ and $\text{Tr}(\mathcal{Z}^A(\sigma^{\text{SA}})^2)$ thus implement this cost function as claimed.

Appendix D: Reading out the Decoherence Functional Elements

While VCH avoids the need to compute the exponentially many $\mathcal{D}(\alpha, \alpha')$'s in order to determine the consistency of a family \mathcal{F} , we do have the ability to efficiently read out any particular $\mathcal{D}(\alpha, \alpha')$ if desired. Figure S.6 shows the circuit that one can use to read out the real and/or imaginary parts of $\mathcal{D}(\alpha, \alpha')$ out for $\alpha \neq \alpha'$. The post-processing is similar to that of the Swap test [2, 3], except that we add a conditional statement.

FIG. S.6. Circuit to read out $\mathcal{D}(\alpha, \alpha')$. The controlled $U(\alpha, \alpha')$ prepares the state $|\alpha\rangle$ on the B registers when the control qubit is in the state $|0\rangle$ and $|\alpha'\rangle$ when the control qubit is in the state $|1\rangle$, so the combination of the Hadamard gate on C and the controlled $U(\alpha, \alpha')$ prepares a superposition of the histories. The z -rotation in the green box is excluded when we calculate the real part of $\mathcal{D}(\alpha, \alpha')$ and included when we calculate the imaginary part. The post processing is described in the text.

When we exclude the z -rotation, conditioned on the control qubit C being measured in the state $|0\rangle$ we perform the Swap test between the A and B registers to get:

$$\begin{aligned} R_0 &= \text{Tr} \left(\sigma^A \left[\frac{1}{2} (|\alpha\rangle + |\alpha'\rangle)(\langle\alpha| + \langle\alpha'|) \right] \right) \\ &= \frac{1}{2} (\mathcal{D}(\alpha, \alpha) + \mathcal{D}(\alpha', \alpha') + \mathcal{D}(\alpha, \alpha') + \mathcal{D}(\alpha', \alpha)) \\ &= \frac{1}{2} (\mathcal{D}(\alpha, \alpha) + \mathcal{D}(\alpha', \alpha')) + \text{Re}(\mathcal{D}(\alpha, \alpha')). \end{aligned} \quad (\text{D1})$$

If we instead condition on C being measured in the state $|1\rangle$ we perform the Swap test between the A and B registers to get:

$$\begin{aligned} R_1 &= \text{Tr} \left(\sigma^A \left[\frac{1}{2} (|\alpha\rangle - |\alpha'\rangle)(\langle\alpha| - \langle\alpha'|) \right] \right) \\ &= \frac{1}{2} (\mathcal{D}(\alpha, \alpha) + \mathcal{D}(\alpha', \alpha')) - \text{Re}(\mathcal{D}(\alpha, \alpha')). \end{aligned} \quad (\text{D2})$$

Our method therefore separates the output based on the result of measuring C, and then performs the usual Swap test post processing on each partition of the output counts to get R_0 and R_1 . Finally, we combine these to get:

$$\text{Re}(\mathcal{D}(\alpha, \alpha')) = \frac{1}{2}(R_0 - R_1). \quad (\text{D3})$$

Instead including that z -rotation gives us

$$\begin{aligned} I_0 &= \text{Tr} \left(\sigma^A \left[\frac{1}{2} (|\alpha\rangle + i|\alpha'\rangle)(\langle\alpha| - i\langle\alpha'|) \right] \right) \\ &= \frac{1}{2} (\mathcal{D}(\alpha, \alpha) + \mathcal{D}(\alpha', \alpha') - i\mathcal{D}(\alpha, \alpha') + i\mathcal{D}(\alpha', \alpha)) \\ &= \frac{1}{2} (\mathcal{D}(\alpha, \alpha) + \mathcal{D}(\alpha', \alpha')) - \text{Im}(\mathcal{D}(\alpha, \alpha')), \end{aligned} \quad (\text{D4})$$

conditioned on C being measured in the state $|0\rangle$. Similarly, conditioned on C being measured in the state $|1\rangle$

we find:

$$\begin{aligned} I_1 &= \text{Tr} \left(\sigma^A \left[\frac{1}{2} (|\alpha\rangle - i|\alpha'\rangle)(\langle\alpha| + i\langle\alpha'|) \right] \right) \\ &= \frac{1}{2} (\mathcal{D}(\alpha, \alpha) + \mathcal{D}(\alpha', \alpha') + i\mathcal{D}(\alpha, \alpha') - i\mathcal{D}(\alpha', \alpha)) \\ &= \frac{1}{2} (\mathcal{D}(\alpha, \alpha) + \mathcal{D}(\alpha', \alpha')) + \text{Im}(\mathcal{D}(\alpha, \alpha')). \end{aligned} \quad (\text{D5})$$

Again, we combine these to get:

$$\text{Im}(\mathcal{D}(\alpha, \alpha')) = \frac{1}{2}(I_1 - I_0) \quad (\text{D6})$$

We also note that the controlled $U(\alpha, \alpha')$ we have made use of here can be implemented with depth that scales linearly in the number of bits by which $|\alpha\rangle$ and $|\alpha'\rangle$ differ. This is accomplished by acting with X gates on all of the registers where the bit-string associated with $|\alpha\rangle$ is 1 followed by CNOT gates from C to each of the registers where the bit-strings for $|\alpha\rangle$ and $|\alpha'\rangle$ differ.

Finally, we comment that reading out $\mathcal{D}(\alpha, \alpha)$ is simpler than the general case as we merely have to prepare $|\alpha\rangle\langle\alpha|$ (which consists of a single layer of X gates) on the B registers and perform the Swap test, without any need for or reference to C.

Appendix E: Implementation Circuits

1. Spin in a Magnetic Field

For our simulations of the spin-1/2 particle in a magnetic field, Fig. S.7 shows the quantum circuit that was used on the simulator and IBM's ibmqx5 processor to perform the cost minimization and to generate the cost landscape plots (shown in Fig. 3).

2. Chiral Molecule

Figure S.8 shows the quantum circuit that was used on a simulator to map the cost function landscapes for the chiral molecule (shown in Fig 4). The tunneling between the chirality states was modeled as a rotation about the z -axis by an angle θ_z . We considered the chiral molecule to be in a gas, and hence its environment is composed of other surrounding molecules that may collide with the molecule of interest. Our model for these collision interactions was implemented by performing a rotation around the x -axis by an angle θ_x (which determines the interaction strength) on an environmental qubit representing the colliding molecule, controlled by the chirality of the molecule of interest.

Appendix F: Highlighted Applications

FIG. S.7. Quantum circuit that we employed to evaluate the cost functions for the spin in a magnetic field. The wires labeled S represent the copies of the spin and those labeled A represent the ancillas. Note that this circuit prepares two copies of σ^A . The gates and measurements inside the solid green box are only included to calculate $\text{Tr}((\sigma^A)^2)$, as without them this is the circuit to calculate $\text{Tr}(\mathcal{Z}^A(\sigma^A)^2)$.

FIG. S.8. Quantum circuit that we employed to evaluate the cost functions for the chiral molecule example in the main text. The wires labeled S represent the chirality degree of freedom of the molecule, E represents the environment (other surrounding molecules), and A represents the ancillas. Note that this circuit prepares two copies of σ^{SA} (and hence σ^A). The gates and measurements inside the blue dashed boxes are only included when we are evaluating the partial trace cost function (i.e., when working with σ^{SA} rather than σ^A). The gates and measurements inside the solid green box are only included when calculating $\text{Tr}((\sigma^A)^2)$ or $\text{Tr}((\sigma^{SA})^2)$, and otherwise the circuit calculates $\text{Tr}(\mathcal{Z}^A(\sigma^A)^2)$ or $\text{Tr}(\mathcal{Z}^A(\sigma^{SA})^2)$.

Here we provide a brief outline of two potential applications that would be viable with NISQ computers.

1. Spin Diffusion

The phenomenon of spin diffusion has been known for a long time [4], but an understanding of the transition from oscillatory dynamics to a classical diffusion equation as system sizes increase is still incomplete. Given that quantum dots with $\sim 30,000$ nuclear spins have been shown to exhibit spin diffusion [5], we expect this to be a very conservative upper bound on the number of spins required.

As a lower bound, simple numerical calculations that we performed show that ~ 10 spins do not appear to exhibit spin diffusion. In particular, our calculations showed that, for these small spin systems, the local magnetization does not provide a consistent family of stationary histories. (See the main text for an example of stationary histories for chiral molecules.) Note that the local magnetization forming a consistent family would be a pre-requisite for a random-walk description (and hence diffusive dynamics) of spin magnetization. Combining this lower bound with our upper bound, we expect that the transition is likely to be found with $\sim 10^2$ or $\sim 10^3$ spins.

Applying VCH to this problem would also illuminate the nature and sharpness of the transition. Namely, we anticipate that the transition will involve the disappearance of Rabi oscillations (a signature of quantum interference) for magnetization as the number of spins increases. A natural question is whether such oscillations disappear completely at a critical system size, analogous to how the chirality oscillations disappeared for the chiral molecule (discussed in the main text) at a particular decoherence rate [6]. Another possibility is that the transition is gradual, rather than sharp, and that the oscillations are merely suppressed rather than eliminated with system size.

It has been experimentally demonstrated with echo techniques that coherence is maintained during spin diffusion [7–9]. In other words, the classical diffusion equation can be understood to arise from closed-system dynamics rather than open-system dynamics, i.e., as an effect of coarse graining rather than an interaction with the environment. We would therefore only be interested in ansatzes that represent coarse grained spin information on some subset of the spins and neglect environmental effects.

Given these considerations, we can estimate the number of qubits needed to apply VCH to this situation and look for the sort of random walks that would give rise to diffusion. Let n_{total} and n_{voxel} respectively denote the total number of spins and the number of spins in the region we are following the magnetization of (the voxel). Simulating n_{total} spins requires n_{total} qubits. In

order to implement the projections, we would need to have at most enough qubits to span a space large enough to account for the $n_{\text{voxel}} + 1$ possible magnetizations, though this could be coarse grained further. Therefore, to carry out this investigation for k times, we would expect to need roughly

$$2(n_{\text{total}} + k \lceil \log_2(n_{\text{voxel}} + 1) \rceil) \quad (\text{F1})$$

qubits, where the factor of two comes from the fact we need two copies of the state for VCH. Thus, our estimate for where we expect to find the transition to diffusive behavior with coarse graining translates to needing somewhere around $\sim 10^2$ or $\sim 10^3$ qubits.

2. Protein Folding

Proteins with up to 76 amino acids have been folded thus far using molecular dynamics simulations without adding in external forces to bias the dynamics towards the "correct" configuration [10]. However, these simulations do not include decoherence effects and are not capable of fully exploring the vast space of un-biased paths. To move beyond what can be done with these classical tools, we propose to use VCH.

In order to investigate under which circumstances a protein will follow a single deterministic path or fold by multiple paths, one could implement a quantum simulation of the process using only realistic interaction Hamiltonians and examine the histories. Conjecturing that decoherence by the environment should play an important role, we would need to consider a simulation of an initially unfolded protein as well as its environment.

Let us consider the simplified case of lattice protein folding for a chain with n_{AA} amino acids. Each connection between amino acids in such a model can be in m different configurations. This system can be represented with $\lceil (n_{\text{AA}} - 1) \log_2(m) \rceil$ qubits. In analogy with the chiral molecule example in the main text, we propose an environment model that would act with different rotations to environment qubits based on the current configuration of each connection, meaning that the size of the environment being modeled would be something like $k \lceil (n_{\text{AA}} - 1) \log_2(m) \rceil$ qubits for k times. The size of the ancillas required to record fine grained histories of this system is the same as this environmental size. Finally, given the need for two copies, we end up with a total qubit requirement of

$$2(2k + 1) \lceil (n_{\text{AA}} - 1) \log_2(m) \rceil \quad (\text{F2})$$

qubits. For a cubic lattice with $n_{\text{AA}} = 100$ examined at 10 times, this becomes 9,660 qubits. Given that such a history analysis becomes classically intractable well before the search for the correct (native) configuration does, we therefore think that useful instances of this application will become practical with quantum computers with between 10^3 and 10^4 qubits.

-
- [1] M. Gell-Mann and J. B. Hartle. Classical equations for quantum systems. *Phys. Rev. D*, 47:3345–3382, 1993.
- [2] J. C. Garcia-Escartin and P. Chamorro-Posada. Swap test and Hong-Ou-Mandel effect are equivalent. *Physical Review A*, 87(5):052330, 2013.
- [3] L. Cincio, Y. Subaşı, A. T. Sornborger, and P. J. Coles. Learning the quantum algorithm for state overlap. *New Journal of Physics*, 20(11):113022, 2018.
- [4] N. Bloembergen. On the interaction of nuclear spins in a crystalline lattice. *Physica*, 15(3):386 – 426, 1949.
- [5] S. Adachi, R. Kaji, S. Furukawa, Y. Yokoyama, and S. Muto. Nuclear spin depolarization via slow spin diffusion in single InAlAs quantum dots observed by using erase-pump-probe technique. *Journal of Applied Physics*, 111(10):103531, 2012.
- [6] P. J. Coles, V. Gheorghiu, and R. B. Griffiths. Consistent histories for tunneling molecules subject to collisional decoherence. *Phys. Rev. A*, 86:042111, 2012.
- [7] Shanmin Zhang, B. H. Meier, and R. R. Ernst. Polarization echoes in nmr. *Phys. Rev. Lett.*, 69:2149–2151, Oct 1992.
- [8] Torgny Karlsson, Michael Helmle, N.D. Kurur, and Malcolm H. Levitt. Rotational resonance echoes in the nuclear magnetic resonance of spinning solids. *Chemical Physics Letters*, 247(4):534 – 540, 1995.
- [9] M. Tomaselli, S. Hediger, D. Suter, and R. R. Ernst. Nuclear magnetic resonance polarization and coherence echoes in static and rotating solids. *The Journal of Chemical Physics*, 105(24):10672–10681, 1996.
- [10] Stefano Piana, Kresten Lindorff-Larsen, and David E. Shaw. Atomic-level description of ubiquitin folding. *Proceedings of the National Academy of Sciences*, 110(15):5915–5920, 2013.

REVIEWERS' COMMENTS:

Reviewer #2 (Remarks to the Author):

I believe with the additional changes and updates this work is suitable for publication in Nature Communications.

“Reviewer #2 (Remarks to the Author):

I believe with the additional changes and updates this work is suitable for publication in Nature Communications.”

We appreciate this evaluation and naturally agree. Thank you.

Sincerely,

Patrick Coles, LANL, pcoles@lanl.gov (Corresponding Author)